# FusionFactory: Fusing LLM Capabilities with Multi-LLM Log Data

**Tao Feng**[*]                                                          *taofeng2@illinois.edu*
*University of Illinois Urbana-Champaign*

**Haozhen Zhang**[*]                                                    *wazhz14@gmail.com*
*Nanyang Technological University*

**Zijie Lei**[*]                                                        *zijielei303@meta.com*
*Meta Monetization AI*

**Pengrui Han**                                                          *phan12@illinois.edu*
*University of Illinois Urbana-Champaign*

**Mostofa Patwary**                                                      *mpatwary@nvidia.com*
*NVIDIA*

**Mohammad Shoeybi**                                                     *mshoeybi@nvidia.com*
*NVIDIA*

**Bryan Catanzaro**                                                      *bcatanzaro@nvidia.com*
*NVIDIA*

**Jiaxuan You**                                                          *jiaxuan@illinois.edu*
*University of Illinois Urbana-Champaign*

**Reviewed on OpenReview:** *https://openreview.net/forum?id=N951scS3yE*

## Abstract

The rapid advancement of large language models (LLMs) has created a diverse landscape of models, each excelling at different tasks. This diversity drives researchers to employ multiple LLMs in practice, leaving behind valuable multi-LLM log data. This naturally leads to the question of whether such logs can be fully leveraged to fuse LLMs' complementary capabilities. Although prior work has explored various strategies for integrating multiple LLMs, we argue that practical fusion must meet two essential requirements: (1) *compatibility with real-world serving scenarios* (*e.g.*, local and API-based serving), and (2) *flexibility to operate at different stages of the LLM pipeline to meet varied user needs* (*e.g.*, fine-tuning and inference stages). To this end, we introduce `LLMFusionBench`, a large-scale benchmark for LLM fusion that spans 14 tasks across six domains, with responses from 20 open-source LLMs (8B–671B) totaling 103M tokens. Building on `LLMFusionBench`, we propose `FusionFactory`, a systematic framework with three elaborated levels: **(1) query-level fusion** via tailored LLM routers, **(2) thought-level fusion** leveraging retrieved abstract reasoning templates, and **(3) model-level fusion** via distillation from top-ranked responses. Experiments show that `FusionFactory` consistently outperforms the best individual LLM across all 14 benchmarks, with the optimal fusion configuration varying across benchmarks, highlighting the promise of multi-LLM log data as a practical foundation for fusing diverse LLM capabilities.

 ulab-uiuc/FusionFactory      🤗 Hugging Face Collection

---

[*] Equal contribution.

# 1 Introduction

Large language models (LLMs), with differences in architecture design, training data, and optimization objectives, often excel at different tasks (Ahmed et al., 2024; Zhang et al., 2024b; Singhal et al., 2022; Luo et al., 2022). For example, DeepSeek-R1 achieves strong results in code generation (Guo et al., 2025), while Claude 3.7 Sonnet is particularly effective in factual retrieval and knowledge coverage (Anthropic, 2025); our work also verifies this diversity of LLM performance (Figure 2 and 3). Due to this diversity, researchers tend to utilize multiple LLMs in their workflow, notably in LLM API server platforms and agentic workflows (Zhang et al., 2024a); such usage pattern leaves us with valuable multi-LLM log data. Here, multi-LLM log data refers to structured serving records produced when multiple LLMs are invoked for user queries, rather than an ordinary collection of independent model responses. Such records preserve not only query-response pairs, but also model identity, task or domain information, response mode, inference cost, and quality feedback, which together characterize the behavior of different LLMs under aligned query settings. We point out that these multi-LLM log data are a valuable vault, leading to an important research question: *can we fully leverage multi-LLM log data to fuse LLMs' complementary capabilities?*

To address this challenge, prior work has explored several different ways of integrating multiple LLMs. Some studies investigate multi-agent systems, where LLMs collaborate through division of labor (Talebirad & Nadiri, 2023; Han et al., 2024c; Chen et al., 2023b). Others develop LLM routers that dynamically assign queries to the most suitable model (Ding et al., 2024a; Dai et al., 2024b; Chen et al., 2023a; Ong et al., 2024; Feng et al., 2024b), while some existing model fusion approaches, including naive ensembles (Haber & Holtzman Gazit, 2013; Li et al., 2023b) and model merging (Yang et al., 2024b), aim to leverage multiple LLMs at the output and parameter levels, respectively. Despite their different formulations, these approaches share the same underlying goal of harnessing complementary strengths among LLMs. However, most existing methods study only one specific fusion stage in isolation. For example, routing methods mainly use multi-LLM records for query-level model selection, while distillation or model-fusion methods mainly focus on transferring responses or knowledge into a target model. As a result, they do not fully answer a practical question faced by users: given the same source of multi-LLM log data, should it be used for routing, thought-level prompting, or model-level distillation? While these explorations mark an important step forward, fundamental challenges remain in building LLM fusion approaches that are both broadly effective and practically feasible. These challenges motivate a closer examination of what properties an effective fusion framework should possess.

In this work, we argue that effective fusion methods should ideally satisfy two key requirements. *First, effective fusion should be compatible with different LLM serving scenarios*: Real-world LLM servicing, broadly speaking, can be categorized into local serving (when model weights are available) and API-based serving (when models are closed-source or expensive to run). While LLM distillation (Xu et al., 2024) has emerged as a popular paradigm for fusion, existing methods typically rely on logits or internal states, which are inaccessible in the generic LLM serving scenarios (*e.g.*, API-based serving). In contrast, multi-LLM log data, which is stored in text format, offers the most practical, cost-effective, and scalable foundation for studying LLM fusion, and thus forms the focus of this work. Different from generic multi-response collections that mainly contain several outputs for the same input, the log data considered in this work contains structured metadata and feedback signals associated with LLM serving, including model identity, response mode, cost, task-specific performance, and LLM-based quality scores. These signals are directly useful for learning routing policies, retrieving reusable reasoning patterns, and selecting high-quality responses for distillation. *Second, effective fusion should happen at different stages in the LLM pipeline to accommodate different users' needs.* In practice, users may or may not have the privilege of finetuning an LLM (due to cost and model availability considerations). Moreover, different tasks may require an LLM to fuse at different stages; for example, a RAG-based LLM system works more favorably with retrieval-based thought-level LLM fusion. In this work, we aim to systematically understand the trade-offs of LLM fusion at different stages. This stage-aware perspective is central to our motivation. API-only compatibility is not the sole novelty of our work; rather, we aim to provide a unified benchmark and framework for studying how the same structured multi-LLM logs can be exploited at different intervention points in the LLM pipeline. This motivates the fusion-stage matrix studied in this paper, which compares query-level, thought-level, and model-level fusion under a shared log source and evaluation protocol.

Table 1: **Comparison of `FusionFactory` with existing works leveraging multi-LLM log data.** We evaluate along three fusion stages, API serving compatibility, and task domain coverage. `FusionFactory` supports all three fusion stages, is compatible with API-based serving, and covers six task domains, demonstrating broader applicability and flexibility compared to prior work.

| Method | LLM Fusion Stage | | | API Serving Compatibility | Task Domain Coverage |
|---|---|---|---|---|---|
| | Query-level | Thought-level | Model-level | | |
| RouteLLM (Ong et al., 2024) | ✓ | ✗ | ✗ | ✓ | 3 |
| GraphRouter (Feng et al., 2024b) | ✓ | ✗ | ✗ | ✓ | 4 |
| RouterBench (Hu et al., 2024) | ✓ | ✗ | ✗ | ✓ | 5 |
| LLM-Blender (Jiang et al., 2023b) | ✓ | ✗ | ✗ | ✓ | 4 |
| FuseLLM (Wan et al., 2024) | ✗ | ✗ | ✓ | ✗ | 3 |
| `FusionFactory` | ✓ | ✓ | ✓ | ✓ | **6** |

As illustrated in Table 1, the unique value of `FusionFactory` is not merely API-serving compatibility, since several prior routing methods also support API-only usage. Instead, `FusionFactory` differs by systematically studying how multi-LLM log data can support multiple fusion stages under a unified framework. Prior routing methods such as RouteLLM, GraphRouter, RouterBench, and LLM-Blender mainly focus on query-level fusion, while FuseLLM mainly studies model-level fusion. In contrast, `FusionFactory` jointly covers query-level, thought-level, and model-level fusion, which enables a cross-stage comparison of when and how the same log data should be used.

Based on the above observations, we propose `LLMFusionBench`, a large-scale benchmark designed to support comprehensive studies on LLM fusion. `LLMFusionBench` provides rich supervision by including both direct and reasoning-augmented responses from multiple LLMs, as well as reusable thought templates summarized from the top-performing models. Covering 14 tasks across 6 domains, `LLMFusionBench` is built from 20 open-source LLMs ranging from 8B to 671B, making it one of the most diverse resources for studying multi-LLM integration (Table 2 and 3). In this sense, `LLMFusionBench` operationalizes the above notion of multi-LLM log data: for each query, it stores aligned responses from multiple LLMs together with their response modes, performance scores, inference costs, and LLM-judge scores. This structure distinguishes `LLMFusionBench` from ordinary multi-response datasets and provides the supervision signals needed for flexible capability fusion.

Building on `LLMFusionBench`, we introduce `FusionFactory`, a systematic framework for LLM fusion across three representative levels (comparison shown in Table 1). `FusionFactory` leverages different components of the structured log records at different fusion stages. Query-level fusion uses query and task information, model identity, performance, cost, and judge scores to learn routing decisions. Thought-level fusion uses historical queries, response modes, and top-ranked reasoning traces to construct reusable thought templates. Model-level fusion uses quality-ranked logged responses as distillation data. **(1) Query-level fusion** adapts to the varying strengths of different models and the diverse user preferences regarding token generation costs by constructing routers tailored to each query, enabling finer-grained capability fusion. Unlike traditional methods, we incorporate both direct outputs and reasoning-augmented responses (Yuan et al., 2024; Besta et al., 2024; Yao et al., 2023; Wei et al., 2022) to better address complex user queries. **(2) Thought-level fusion** improves new responses by retrieving abstract thought templates (*i.e.*, concise summaries distilled from top-performing model outputs) based on embedding similarity to past queries. This design avoids the noise of full past responses while still capturing useful reasoning patterns, which are then used as few-shot demonstrations for new queries (Song et al., 2023; Kang et al., 2023; Li et al., 2024b; Zhao et al., 2021). **(3) Model-level fusion** takes a distillation view (Sreenivas et al., 2024; Stichlmair et al., 2021), transferring complementary capabilities into a single base model by conducting supervised fine-tuning on high-quality responses aggregated across LLMs. Inspired by imitation learning in the LLM distillation domain (Hussein et al., 2017; Osa et al., 2018), for each query, we select the LLM responses with the top-k performance or the highest LLM judge scores as training data. This enables downstream deployment of a fused model without relying on multiple LLMs at inference time.

We systematically evaluate `FusionFactory` on `LLMFusionBench` and find that *fusing LLMs with `FusionFactory` yields consistent benchmark performance improvement over the best individual LLM across 14 popular LLM benchmarks*, where the optimal `FusionFactory` configuration varies across different benchmarks. Specifically, we find that: (1) *thought-level fusion achieves the best overall performance*, especially

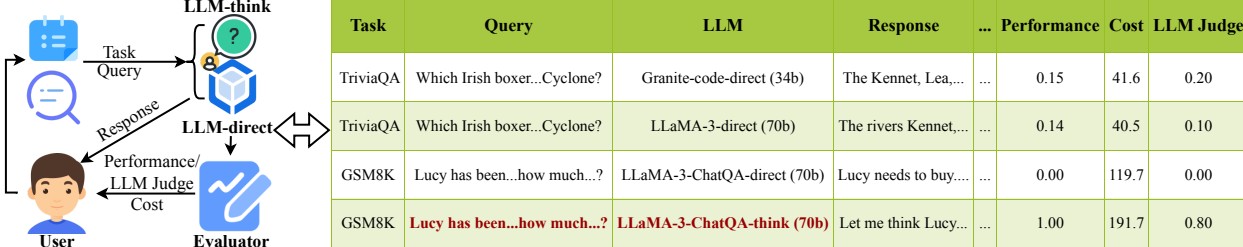

| Task | Query | LLM | Response | ... | Performance | Cost | LLM Judge |
|------|-------|-----|----------|-----|-------------|------|-----------|
| TriviaQA | Which Irish boxer...Cyclone? | Granite-code-direct (34b) | The Kennet, Lea,... | ... | 0.15 | 41.6 | 0.20 |
| TriviaQA | Which Irish boxer...Cyclone? | LLaMA-3-direct (70b) | The rivers Kennet,... | ... | 0.14 | 40.5 | 0.10 |
| GSM8K | Lucy has been...how much...? | LLaMA-3-ChatQA-direct (70b) | Lucy needs to buy.... | ... | 0.00 | 119.7 | 0.00 |
| GSM8K | **Lucy has been...how much...?** | **LLaMA-3-ChatQA-think (70b)** | Let me think Lucy... | ... | 1.00 | 191.7 | 0.80 |

Figure 1: **Overview of `LLMFusionBench`'s construction process.** For each task and user query, multiple LLMs generate responses in two modes: direct and reason (*i.e.*, chain-of-thoughts (CoT) (Wei et al., 2022)). These responses are then evaluated for performance, cost, and judge scores.

Table 2: **`LLMFusionBench` covers 14 tasks across 6 representative domains.**

| Domain | Tasks |
|--------|-------|
| Math | GSM8K, MATH |
| Code | MBPP, HumanEval |
| Commonsense Reasoning | CommonsenseQA, ARC, HellaSwag, OpenBookQA |
| World Knowledge | NaturalQuestions, TriviaQA |
| Reading Comprehension | SQuAD, BoolQ |
| Popular | MMLU, GPQA |

Table 3: **`LLMFusionBench` includes 20 LLMs grouped into three size-based scales.**

| Scale | LLMs |
|-------|------|
| Small | Qwen2 (7b), Mistral (7b), Qwen2.5 (7b), Gemma (7b), CodeGemma (7b), Gemma-2 (9b), LLaMA-3.1 (8b), Granite (8b), LLaMA-3 ChatQA (8b), Mistral-Nemo (12b) |
| Medium | LLaMA-3.3 Nemotron Super (49b), Granite Code (34b), LLaMA-3.1 Nemotron (51b), LLaMA-3 ChatQA (70b), LLaMA-3.1 (70b), LLaMA-3 (70b), Mixtral (8x7b) |
| Large | Palmyra Creative (122b), Mixtral (8x22b), DeepSeek-R1 (671b) |

when using hybrid selection and large summaries; (2) *model-level fusion performs the worst among the three levels*, likely due to overfitting and difficulty generalizing across tasks; (3) *query-level fusion offers a good balance between performance and efficiency*, achieving decent results while requiring minimal computational overhead; (4) *gains from fusing World Knowledge and Math domains are moderate*, where these domains demand high factual accuracy or strict logical consistency, which are difficult to maintain through multi-LLM fusion. These results highlight the promise of multi-LLM log data as a practical foundation for developing fusion methods that flexibly adapt to diverse serving scenarios and fusion stages. Moreover, the cross-level comparison enabled by our fusion-stage matrix provides insights that cannot be obtained by studying routing or distillation alone. It reveals that different ways of using the same multi-LLM log data lead to different performance, cost, and generalization behaviors, thereby providing a practical decision framework for choosing among routing-based, thought-based, and training-based fusion under different deployment constraints.

## 2 `LLMFusionBench`: Benchmarking LLM Capability Fusion with Multi-LLM Log Data

In this section, we detail the construction of `LLMFusionBench` (Section 2.1) and its support for LLM capability fusion (Section 2.2), as illustrated in Figure 1.

### 2.1 `LLMFusionBench` Construction

As shown in Figure 1, we construct `LLMFusionBench` by sampling up to 500 training and 50 partial test queries (10:1 ratio) from each of 14 tasks across six domains (Table 2). For each query, responses from multiple LLMs are collected to support comparison and exploration of query-, thought-, and model-level fusion (Sections 4.2, 5.2, and 6.2). For the final cross-level analysis (Section 7), we also use each task's full test set to comprehensively evaluate fusion levels. Table 10 in the Appendix summarizes sample counts for training, partial test, and full test sets.

We formalize the resulting multi-LLM log data as structured serving records rather than ordinary response collections. Specifically, each log entry is represented as $(q, t, m, s, r, c, y, j)$, where $q$ denotes the user query, $t$ denotes the task or domain, $m$ denotes the LLM identity, $s$ denotes the response mode such as direct or reasoning-augmented prompting, $r$ denotes the generated response, $c$ denotes the inference cost, $y$ denotes the

| | |
|---|---|
| **\<Query\>** What is the positive difference between $(6^2 + 6^2)/6$ and $(6^2 \times 6^2)/6$? | **Response of LLaMA-3-think (70b)**: Step 1: Evaluate each expression separately. |
| **Response of LLaMA-3.1 Nemotron-direct (51b)**: 204. | Step 2: The first expression is $(6^2 + 6^2)/6$, and the second expression is $(6^2 \times 6^2)/6$. |
| **Response of LLaMA-3.1 Nemotron-think (51b)**: To find the positive difference | Step 3: We have $6^2 = 36$, so the expression becomes $(36 + 36)/6$. |
| between $(6^2+6^2)/6$ and $(6^2 \times 6^2)/6$, we need to evaluate each expression separately. | Step 4: Adding the numbers in the numerator, we get $72/6$. |
| First, let's evaluate $(6^2+6^2)/6$: $(6^2+6^2)/6 = (36+36)/6 = 72/6 = 12$ | Step 5: Dividing 72 by 6 gives us 12. |
| Next, let's evaluate $(6^2 \times 6^2)/6$: $(6^2 \times 6^2)/6 = (36 \times 36)/6 = 1296/6 = 216$ | Step 6: Then we have $6^2 = 36$, so the expression becomes $(36 \times 36)/6$. |
| Now, we need to find the positive difference between 12 and 216: $216 - 12 = 204$ | Step 7: Multiplying 36 by 36 gives us 1296, and then dividing by 6 gives us 216. |
| So, the positive difference between $(6^2+6^2)/6$ and $(6^2 \times 6^2)/6$ is 204. | Step 8: The positive difference is $216 - 12 = 204$.The final answer is: 204 |

Figure 2: **LLMs exhibit varying reasoning styles when answering the same query.** This example from the Math task compares responses from LLaMA-3.1 Nemotron-direct (51B), Nemotron-think (51B), and LLaMA-3-think (70B), where -think denotes CoT prompting. The responses show a progression from concise answers to increasingly detailed reasoning.

task-specific evaluation result, and $j$ denotes the LLM-judge score. Under this definition, `LLMFusionBench` is not simply a set of $(q, r)$ pairs, but an aligned multi-LLM serving log that records model behavior, response style, cost, and quality feedback for the same query.

As summarized in Table 3, we select 20 LLMs across varying scales. To elicit diverse responses, we apply distinct system prompts for direct and reasoning-based outputs. The 20-model pool contains both LLMs with native trained think-token behavior (e.g., DeepSeek-R1 and the Nemotron `-think` variants) and LLMs without it. We apply the same chain-of-thought elicitation procedure uniformly across all 20 models to enable apples-to-apples comparison across the pool; for models with trained think-token behavior the model's native reasoning style is preserved when responding to our prompt, while for models without it the `-think` responses are produced via prompted CoT. Responses are evaluated with task-specific metrics (Table 7), but as shown in Figure 17, such metrics may favor short, low-information answers that hinder effective fusion. To address this, we introduce an LLM judge that scores the insightfulness of correct answers (prompt in Table 16). As shown in Table 17, the LLM judge better reflects informative value, offering greater utility for fusion.

To assess cost, we follow (Feng et al., 2024b) by estimating total token usage via GPT-2 (Radford et al., 2019) and converting it into monetary cost using Together AI's API pricing (Table 8). As shown in the right of Figure 1, collected data is stored in a DataFrame, with each row capturing task, query, LLM, response, performance, cost, and LLM judge score (details in Table 9).

This construction focuses on a practical subset of log signals that can be consistently collected in both local and API-based LLM serving scenarios. Richer production logs may additionally contain temporal order, explicit user feedback, error patterns, and multi-turn interaction history. These signals are valuable, but they are not always available across different serving settings, especially when using closed-source or API-based models. Therefore, `LLMFusionBench` emphasizes broadly accessible log attributes, including model identity, response mode, cost, task-specific performance, and LLM-based quality feedback, while leaving interaction-level signals such as temporal dynamics and user feedback as important extensions.

## 2.2 Support for LLM Capabilities Fusion

As introduced earlier, `LLMFusionBench` captures diverse LLM responses to the same query, enabling potential for LLM capability fusion. To illustrate this, we conduct a qualitative case study using a Math query (Figure 2). We compare responses from LLaMA-3.1 Nemotron-direct (51B), LLaMA-3.1 Nemotron-think (51B), and LLaMA-3-think (70B), where -think denotes CoT prompting. These responses exhibit distinct styles: direct and concise, analytical reasoning, and detailed explanation, respectively. These variations highlight opportunities to learn diverse reasoning patterns and enhance fusion effectiveness.

The structured log format is crucial for this capability-fusion analysis. Because different LLMs are evaluated on the same query with shared task-specific metrics, cost estimates, and judge scores, `LLMFusionBench`

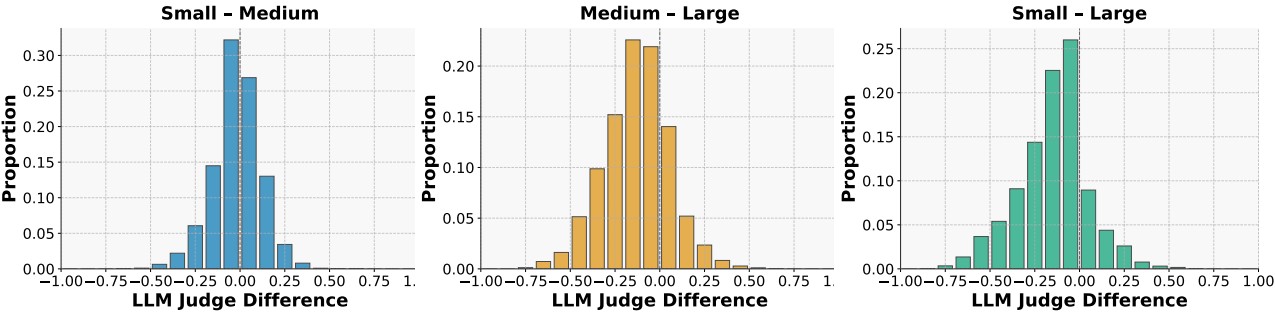

Figure 3: **LLMs of different scales show distinct strengths across queries.** For each query, we compute average LLM judge scores for small, medium, and large models (Table 3), and subtract these scores pairwise (small–medium, medium–large, small–large) to measure relative advantages. We then visualize the resulting score differences as three histograms.

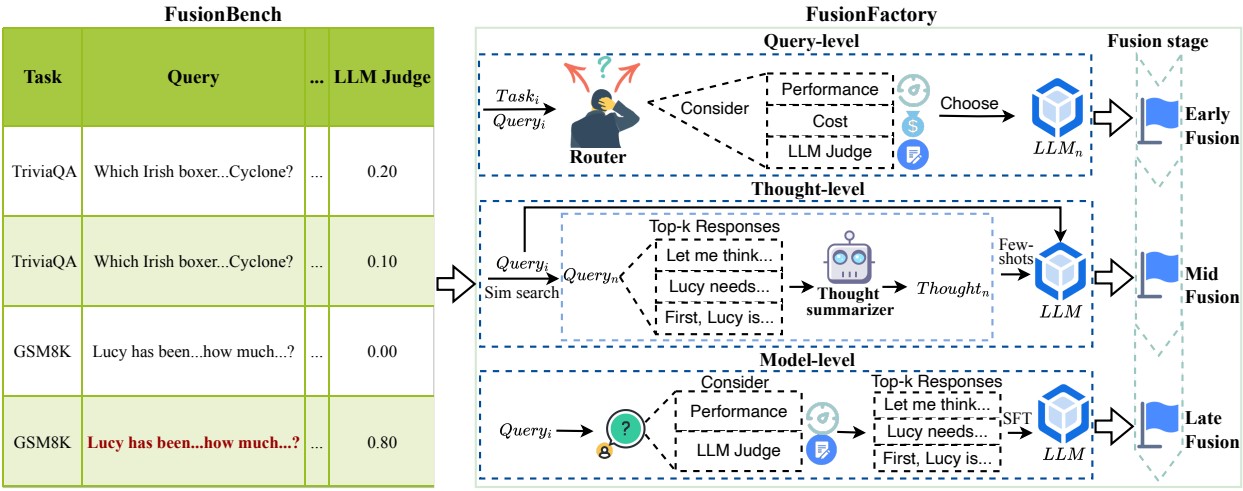

Figure 4: **Overview of `FusionFactory` via `LLMFusionBench` across three fusion stages:** *query-*, *thought-*, and *model-level*, corresponding to early, mid, and late fusion. At the query level, a router selects the best LLM per query using performance, cost, and judge scores. At the thought level, a summarizer extracts thought templates from top-k responses for few-shot prompting. At the model level, we fine-tune a base LLM on high-quality responses to encode fused capabilities.

allows us to compare aligned model behaviors rather than unrelated responses. Model identity helps identify which LLM is suitable for which type of query, response mode distinguishes direct answering from reasoning-augmented generation, and performance or judge feedback provides supervision for selecting useful responses. These log properties are later used by query-level fusion for routing, thought-level fusion for retrieving and summarizing reasoning templates, and model-level fusion for selecting distillation data.

To complement our analysis, we conduct quantitative evaluations using LLM judge scores. As shown in Figure 3, we compute the average judge score for each model scale per query (based on Table 3), and then calculate pairwise differences: small-medium, medium-large, and small-large. The resulting histograms reveal distinct performance gaps across scales, suggesting that models of different sizes exhibit complementary strengths and further motivating fusion via `LLMFusionBench`.

## 3   `FusionFactory`: Flexible Fusion of LLM Capabilities

As shown in Figure 4, we present `FusionFactory` across three representative levels of LLM capabilities fusion: query-, thought-, and model-level, corresponding to early, mid, and late fusion stages. Each level targets a distinct phase in the reasoning process.

This stage-aware organization provides the conceptual taxonomy used in this empirical study. We do not aim to provide formal theoretical guarantees or closed-form complexity bounds; instead, our goal is to systematically compare practical fusion strategies according to where they intervene in the LLM pipeline. Query-level fusion intervenes before generation by selecting a model or response mode for each query. Thought-level fusion intervenes during generation by retrieving and injecting reusable reasoning templates. Model-level fusion intervenes after response collection by consolidating selected logged responses into model parameters through supervised fine-tuning. This taxonomy clarifies that the three levels are not arbitrary engineering categories, but distinct intervention points in the LLM inference and learning process.

This stage-aware organization is also the basis of the fusion-stage matrix studied in this paper. Rather than treating API-only compatibility as the main novelty, `FusionFactory` aims to answer a broader question: given the same structured multi-LLM log data, should it be used before generation for routing, during generation for reasoning augmentation, or after response collection for model training? As illustrated in Figure 4, query-level fusion is an early-stage intervention before response generation, thought-level fusion is a mid-stage intervention during reasoning, and model-level fusion is a late-stage intervention that consolidates logged responses into model parameters. This matrix therefore provides a unified way to compare different uses of multi-LLM logs under the same benchmark and evaluation protocol.

Importantly, the three levels leverage different components of the structured multi-LLM log records introduced in Section 2.1. Query-level fusion uses query and task information, model identity, response mode, performance, cost, and LLM-judge scores to learn routing decisions. Thought-level fusion uses historical queries, response modes, generated responses, and quality-ranked reasoning traces to construct reusable thought templates. Model-level fusion uses task-specific performance or LLM-judge feedback to select high-quality logged responses as supervised fine-tuning data. Therefore, `FusionFactory` does not treat multi-LLM logs as ordinary text-only response collections; instead, it exploits their structured metadata and feedback signals to support different forms of capability fusion.

**Query-level Fusion (Early Fusion)** (Section 4) operates at the problem interpretation stage. A router analyzes the query and task characteristics to select the best LLM configuration, considering performance, cost, and LLM-based evaluations. The selected model then processes the query independently, enabling efficient task-specialized execution. **Thought-level Fusion (Mid Fusion)** (Section 5) intervenes during reasoning. An LLM summarizer distills abstract thought templates from top-k responses (ranked by performance or LLM judgment). For new queries, we retrieve similar historical queries and apply their thought templates as few-shot prompts, allowing dynamic coordination across reasoning styles. **Model-level Fusion (Late Fusion)** (Section 6) occurs after full responses are generated. A base LLM is fine-tuned using top-k high-quality outputs (selected via performance or LLM judgment) to internalize fused capabilities. This stage reflects post-hoc fusion via training rather than online collaboration. **These three fusion stages offer complementary strengths at different points in the reasoning pipeline, allowing users to flexibly choose the most suitable level of integration based on their specific serving constraints, performance goals, and resource availability.**

From a practical computational and deployment perspective, the three stages expose different trade-offs. Query-level fusion has the lowest inference overhead because it only requires a lightweight routing step before calling one selected LLM. Thought-level fusion introduces additional retrieval and prompt-token overhead, but it avoids model fine-tuning and remains compatible with API-based LLMs. Model-level fusion requires offline training and access to a trainable base model, but after training it only requires a single fused model at inference time. We present these as practical deployment trade-offs rather than formal complexity bounds, because exact theoretical costs are difficult to define uniformly across heterogeneous LLM APIs, model sizes, serving infrastructures, and pricing schemes.

This comparison is necessary because existing routing and distillation methods typically focus on one stage in isolation. Routing-based methods mainly study query-level model selection, while distillation-based methods mainly study model-level capability transfer. In contrast, `FusionFactory` uses the same multi-LLM log source to study all three stages together. This enables cross-stage analysis of performance, cost, and generalization behavior, and provides a practical decision framework for users who face different deployment constraints.

## 4 Query-level `FusionFactory` Fusion

### 4.1 Methodology: Query-level Fusion

The goal of query-level fusion is to assign the most suitable LLM for each user query, fusing LLMs' capabilities through a router. This router selects models based on the query, task info, and LLM features, jointly considering performance, cost, and LLM judge scores. Under the structured log formulation in Section 2.1, this level directly uses the log fields related to the input and serving decision, including the query $q$, task or domain $t$, model identity $m$, response mode $s$, cost $c$, task-specific performance $y$, and LLM-judge score $j$. To clarify the routing notation, we define each router action as $a = (m, s) \in \mathcal{A}$, where $m$ denotes the selected LLM and $s$ denotes the selected response mode, such as direct or reasoning-augmented prompting. Thus, the router chooses an LLM-response-mode configuration rather than an unspecified model feature. As shown in the first panel of Figure 4, we follow Feng et al. (2024b) and define the reward for selecting action $a$ for query-task pair $(q, t)$ as:

$$\text{Reward}(q, t, a) = \alpha \cdot P(q, t, a) - \beta \cdot C(q, t, a) + \gamma \cdot J(q, t, a), \tag{1}$$

where $P(q, t, a)$ denotes the task-specific performance score obtained by applying the evaluation metric of task $t$ to the response generated under action $a$; $C(q, t, a)$ denotes the inference cost, estimated from token consumption and API pricing; and $J(q, t, a)$ denotes the LLM-judge score, which measures the informativeness or usefulness of the generated response beyond the task-specific metric. The coefficients $\alpha$, $\beta$, and $\gamma$ control the relative importance of task performance, inference cost, and LLM-judge quality, respectively. The negative sign before $C(q, t, a)$ reflects that lower inference cost is preferred, while higher performance and judge scores are preferred. Based on this, for a query $q$ and its corresponding task $t$, our goal is to learn a router model $f_\phi$ such as a neural network that maps $(q, t)$ to an action $a \in \mathcal{A}$ and maximizes the expected reward:

$$\phi^* = \arg\max_\phi \ \mathbb{E}_{(q,t) \sim \mathcal{D}} \left[ \text{Reward} \left( q, t, f_\phi(q, t) \right) \right], \tag{2}$$

where $\mathcal{D}$ denotes the distribution of query-task pairs and $\mathcal{A}$ denotes the set of candidate LLM-response-mode configurations. After obtaining $\phi^*$, $f_{\phi^*}$ selects the action $a = f_{\phi^*}(q, t)$ for each new query-task pair, and the selected LLM configuration generates the final response. This revised formulation removes the ambiguous dot notation in the original reward function and explicitly connects the routing objective to the structured log fields used in query-level fusion.

From the perspective of the fusion-stage matrix, query-level fusion offers a lightweight way to exploit multi-LLM logs in API-based serving settings. It does not require access to model weights, logits, or internal states; instead, it only requires logged performance, cost, and judge signals to learn a routing policy. Thus, this stage is suitable when users want to improve the accuracy–cost trade-off while still calling only one selected LLM at inference time.

### 4.2 Experimental Settings: Query-level Fusion

**Scenario**. To explore query-level fusion, we define four routing scenarios aligned with Section 4: *Performance First*, *Balance*, *Cost First*, and *LLM Judge*. The first three examine trade-offs between performance and cost, while the last assesses routing under LLM-as-judge evaluation. These scenarios differ in their reward weights $(\alpha, \beta, \gamma)$ in Equation 1, which control the emphasis on performance, cost, and LLM-judge scores and are set to $(1.0, 0.0, 0.0)$, $(0.5, 0.5, 0.0)$, $(0.2, 0.8, 0.0)$, and $(0.0, 0.0, 1.0)$, respectively.

Table 4: **Multiple query-level fusion methods surpass the best single LLM across four scenarios. Bold** and underline indicate the best and second-best scores. In the Performance First, Balance, and Cost First settings, we report Performance, Cost, and Reward to evaluate how well each router balances performance and cost. The LLM Judge scenario evaluates router ability to integrate informative responses, using LLMScore and Performance.

| Scenario | Performance First | | | Balance | | | Cost First | | | LLM Judge | |
|---|---|---|---|---|---|---|---|---|---|---|---|
| Method | Performance | Cost | Reward | Performance | Cost | Reward | Performance | Cost | Reward | LLMScore | Performance |
| Best LLM | 0.546 | 0.1850 | 0.546 | 0.556 | 0.0681 | 0.244 | 0.431 | 0.0047 | 0.0823 | 0.823 | 0.431 |
| RouterKNN | 0.558 | 0.0666 | 0.558 | 0.461 | 0.0068 | 0.227 | 0.463 | 0.0067 | 0.0872 | 0.835 | 0.470 |
| RouterSVM | 0.546 | 0.0673 | 0.546 | 0.448 | 0.0084 | 0.220 | 0.442 | 0.0075 | 0.0825 | 0.790 | 0.449 |
| RouterMLP | 0.561 | 0.0652 | 0.561 | 0.482 | 0.0095 | 0.236 | 0.477 | 0.0078 | 0.0884 | 0.836 | 0.464 |
| RouterBERT | 0.528 | 0.1030 | 0.528 | 0.387 | 0.0262 | 0.180 | 0.368 | 0.0531 | 0.0310 | 0.805 | 0.414 |
| GraphRouter | 0.602 | 0.0184 | **0.602** | 0.464 | 0.0184 | **0.269** | 0.496 | 0.0047 | **0.0955** | **0.870** | **0.538** |

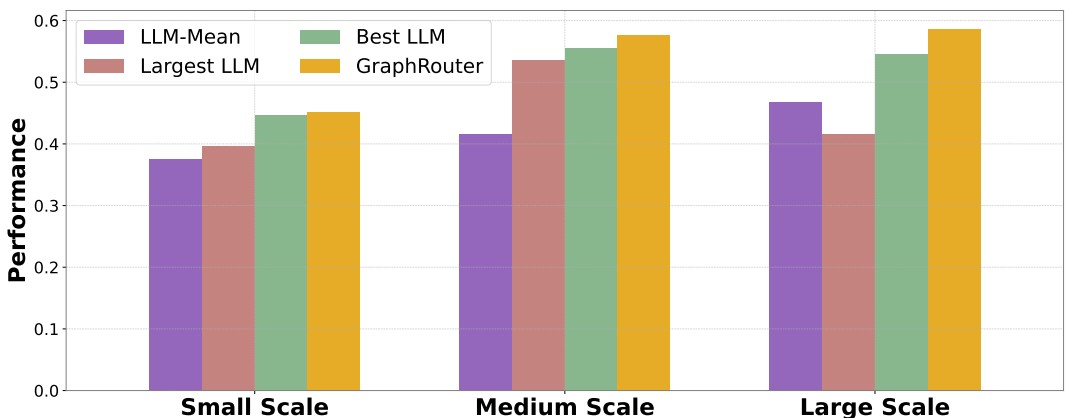

Figure 5: **GraphRouter fuses smaller LLMs to beat both top single and largest models.** Comparison across scales shows larger LLMs don't always perform best under the 1024-token constraint.

**Comparison Routers**. We evaluate five representative routing methods for LLM fusion: *RouterKNN* (Shnitzer et al., 2023), *RouterSVM* (Hu et al., 2024), *RouterMLP* (Shnitzer et al., 2023), *RouterBERT* (Ong et al., 2024), and *GraphRouter* (Feng et al., 2024b). We also include several static baselines: *Best LLM*, which achieves the highest performance on the training set; *LLM-Mean*, which yields the average test performance across all LLMs; and *Largest LLM*, which denotes the test performance of the largest model. Detailed baseline description can be found in Appendix C.

**Evaluation Metric**. Following (Feng et al., 2024b; Ong et al., 2024), we adopt three metrics: *Performance*, the average task accuracy over queries; *Cost*, the average inference expense of selected LLMs; and *LLMScore*, a learned evaluator's quality rating over ⟨query, ground truth, response⟩ triples. Full LLMScore details are in Appendix I.

### 4.3 Result Analysis: Query-level Fusion

We report results in Table 4 with three key findings: **(1) Query-level fusion outperforms the best single LLM across all four scenarios.** As shown in Table 4, methods such as RouterMLP, RouterKNN, and RouterSVM consistently surpass the best individual LLM, with gains of at least 2% and up to 16% in performance or reward, depending on the scenario and metric. **(2) GraphRouter achieves the strongest performance consistently.** It brings over 10% relative reward gains in the performance first and balance settings, 16%+ in cost first, and a 5.7% LLMScore improvement under the LLM Judge scenario. Other routers are less stable, sometimes even underperforming the Best LLM baseline, showing the challenge of effective multi-task multi-LLM fusion. **(3) GraphRouter effectively fuses smaller LLMs to outperform both the best and largest single models.** We further assess GraphRouter's ability across LLM scales (Table 3). Under the performance-first setting, it consistently beats static baselines across small,

medium, and large LLMs (Figure 5), highlighting both the strength of the method and the versatility of `LLMFusionBench` in LLM fusion.

## 5 Thought-level `FusionFactory` Fusion

### 5.1 Methodology: Thought-level Fusion

At the thought level, the core insight is to summarize the responses of multiple LLMs for each query into a thought template, thereby enabling the fusion of different LLMs' capabilities to enhance few-shot responses. This level relies on the log fields that describe historical reasoning behavior, including historical query $q_i$, response mode $s$, generated response $r$, task-specific performance $y$, and LLM-judge score $j$. These fields allow `FusionFactory` to select informative reasoning traces from aligned multi-LLM responses and abstract them into reusable templates, rather than directly retrieving arbitrary raw responses. Specifically, as shown in the second part on the right side of Figure 4, an LLM is employed to serve as a thought summarizer $L_s$, distilling a generalized reasoning template $T_i$ from the top-k responses $R_i$ of query $q_i$ selected according to performance or LLM judge:

$$R_i = \{r_1^i, r_2^i, \ldots, r_k^i\} = \text{Top-}k\left(\{r_j^i\}_{j=1}^n \text{Score}(r_j^i)\right), \qquad T_i = L_s(R_i) \tag{3}$$

where $\{r_j^i\}_{j=1}^n$ denotes the set of responses generated by $n$ different LLMs for query $q^i$, and $\text{Score}(r_j^i)$ is given by task-specific metrics or an LLM judge.

Based on this, for new queries $q^l$, we retrieve the top-d similar queries $\mathcal{Q}^l = (q_1, q_2, ..., q_d)$ via similarity search and use their corresponding thought templates $\mathcal{T}^l = (T_1, T_2, ..., T_d)$ to enable few-shot prompting of the LLM response $R^l$:

$$\mathcal{Q}^l = \text{Top-}d\left(\mathcal{Q}, \text{sim}(q^l, q_i)\right), \qquad \mathcal{T}^l = (T_1, T_2, \ldots, T_d), \qquad R^l = \text{LLM}\left(q^l, \mathcal{T}^l\right) \tag{4}$$

Within the fusion-stage matrix, thought-level fusion occupies a distinct middle stage between routing and distillation. Unlike query-level fusion, it does not merely choose one model before generation. Unlike model-level fusion, it does not require fine-tuning a base model. Instead, it reuses high-quality reasoning traces from the logs as non-parametric guidance at inference time. This makes thought-level fusion useful when users want to exploit historical multi-LLM reasoning patterns while remaining compatible with API-based LLMs and avoiding offline model training.

### 5.2 Experimental Settings: Thought-level Fusion

**Model and Implementation**. We evaluate thought-level fusion across three model sizes and six task domains in `LLMFusionBench`. For each training query, `LLaMA-3 (70B)` generates a high-level *thought template* by summarizing the top-3 responses, selected by *Performance* and *LLMScore*, with cost as tiebreaker (prompt in Table 12). To assess effectiveness, we compare three paradigms: (1) *Zero-Shot*, using only the query; (2) *Few-Shot*, retrieving three similar queries (by cosine similarity) each with three top responses; (3) *Thought Template*, using the same three similar queries, but with distilled templates. For *Thought Template*, we try three strategies: (a) *Performance-based*, using top-3 responses by *Performance*; (b) *LLMScore-based*, by *LLMScore*; (c) *Hybrid*, selecting top-5 by *Performance*, then top-3 by *LLMScore*. We experiment with *llama3-8b* (small), *llama-3.1-nemotron-51b-instruct* (medium), and *palmyra-creative-122b* (large), chosen as top performers in their scale. Accuracy is averaged by domain and shown as radar plots in Figure 6.

### 5.3 Result Analysis: Thought-level Fusion

As shown in Figure 6, thought-level fusion consistently improves performance across all model sizes. Averaged over six task domains, small, medium, and large models improve by 12.7%, 4.8%, and 8.3%, respectively, compared to zero-shot. These improvements manifest differently across reasoning types and model sizes, as detailed below. **(1) Strong gains on reasoning-intensive tasks (relative to zero-shot).** Thought fusion is most effective on tasks like math and code. Measured against the zero-shot baseline, Hybrid fusion

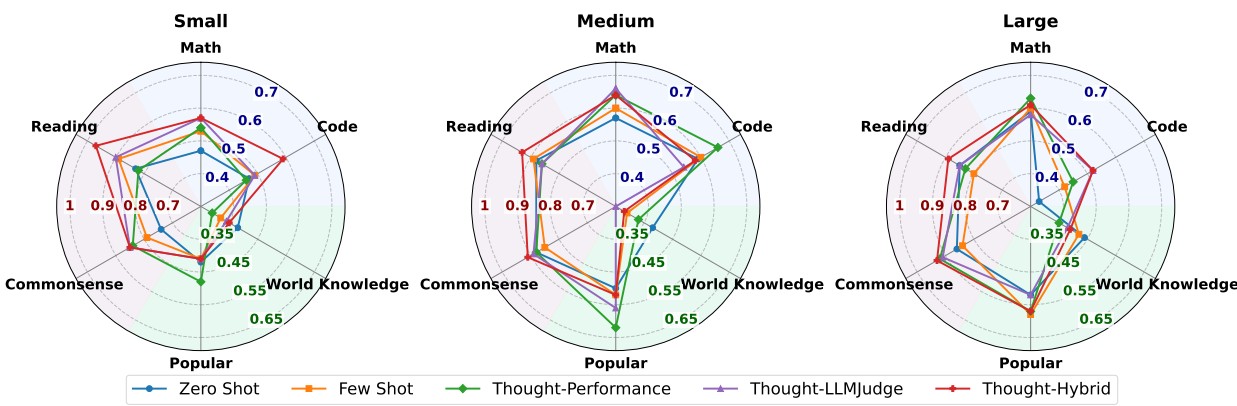

Figure 6: **Thought-level fusion improves performance across model scales and task domains.** Radar plots show accuracy per domain for small (LLaMA-3.1 8B), medium (LLaMA-3.1 Nemotron 51B), and large (Palmyra Creative 122B) models under different paradigms. The thought template approach yields the largest gains, especially on reasoning-heavy tasks like Math and Code. The hybrid strategy balances performance and response quality, often achieving the best results.

improves math by 21.3% (small) and 15.8% (medium); the large model (Palmyra Creative 122B) gains 57.6% in coding (from 0.33 to 0.52). Commonsense and reading comprehension also improve (14.9% and 16.9% for small models). In contrast, performance drops on world knowledge (-7.9% for small models) and varies on popular knowledge, likely due to these tasks emphasizing factual recall over reasoning. Relative to the stronger few-shot baseline (also shown in Figure 6), the best thought-fusion variant remains positive on math and code but the margin is smaller (within ∼5% on several scale–strategy combinations); we therefore present thought-template fusion as a robust alternative to few-shot rather than a uniform improvement over it. **(2) Large models still benefit.** Thought-level fusion yields notable improvements even for large models: +57.6% in code and +6.8% in math. Medium models gain 11.9% (code) and 15.8% (math), showing the value of distilling diverse reasoning even at large model scales. **(3) Hybrid strategy offers best trade-off.** Here, *accuracy* refers to task-specific Performance (e.g., exact-match for QA, pass@1 for code) and *quality* refers to the LLM-judge score (LLMScore) over ⟨query, ground-truth, response⟩ triples (Appendix I); Hybrid selection picks top-5 by Performance, then top-3 by LLMScore as tie-breaker (Section 5.2). Averaged across the six domains for small models, Hybrid achieves the largest mean gain over zero-shot (+12.7%, vs Performance-based +3.6% and LLMScore-based +7.3%); at task granularity, however, no single strategy strictly dominates across all tasks in Figure 6, and the three are best understood as competitive alternatives whose relative ranking depends on the task. The corresponding overall (across-strategy) thought-fusion means for medium and large scales are +4.8% and +8.3% as reported in the opening paragraph of this section, with the per-strategy breakdown for those scales shown visually in Figure 6. See Appendix D and G for analysis of inference time and benefits of reasoning-augmented responses in thought-level fusion.

## 6 Model-level `FusionFactory` Fusion

### 6.1 Methodology: Model-level Fusion

The model-level aims to leverage different LLMs' high-quality responses as training data to fuse LLM capabilities. In this level, the structured logs provide supervised distillation data by pairing each query $q_i$ with quality-ranked responses $R_i$, where the ranking is determined by task-specific performance $y$ or LLM-judge score $j$. Thus, model-level fusion uses the feedback signals stored in the logs to select which responses should be internalized by the base model. Specifically, as shown in the last part on the right side of Figure 4, we obtain the top-k responses $R_i$ of query $q_i$ selected according to performance or LLM judge as equation (3). Based on this, we can obtain a training data containing $n$ samples $Z = \{q_i, R_i\}_{i=1}^{n}$. Finally, we leverage $Z$

Table 5: **Model-level fusion performance of LLaMA-3.1 (8B) on `LLMFusionBench` under different fine-tuning and selection strategies.** We compare Zero-Shot, Top-$K$ Label-only SFT, and Top-$K$ SFT across three response selection criteria: Task Performance, LLM Judge, and Hybrid. Results span six tasks, with **bold** and underline indicating the best and second-best. CS, WK, and Read denote Commonsense, World Knowledge, and Reading Comprehension.

| Sort Criterion | Task Performance | | | | | | LLM Judge | | | | | | Hybrid | | | | | |
|---|---|---|---|---|---|---|---|---|---|---|---|---|---|---|---|---|---|---|
| Scenarios | Math | Code | CS | WK | Read | Popular | Math | Code | CS | WK | Read | Popular | Math | Code | CS | WK | Read | Popular |
| Zero-Shot | 0.473 | **0.564** | 0.725 | 0.300 | 0.703 | 0.415 | 0.473 | 0.564 | 0.725 | **0.300** | **0.703** | 0.415 | 0.473 | 0.564 | 0.725 | **0.300** | 0.703 | 0.415 |
| Top-5 Label-only SFT | 0.217 | 0.510 | 0.785 | 0.150 | 0.646 | 0.372 | 0.217 | 0.510 | 0.785 | 0.150 | 0.646 | 0.372 | 0.217 | 0.510 | 0.785 | 0.150 | 0.646 | 0.372 |
| Top-5 SFT | **0.547** | 0.503 | 0.755 | 0.290 | **0.731** | 0.468 | **0.528** | **0.584** | 0.780 | 0.280 | 0.662 | **0.489** | 0.443 | **0.578** | 0.790 | **0.300** | **0.728** | 0.457 |
| Top-10 Label-only SFT | 0.340 | 0.498 | **0.830** | 0.150 | 0.667 | **0.489** | 0.340 | 0.498 | **0.830** | 0.150 | 0.667 | 0.489 | 0.340 | 0.498 | **0.830** | 0.150 | 0.667 | 0.489 |
| Top-10 SFT | 0.509 | 0.535 | 0.765 | **0.310** | 0.707 | 0.425 | **0.528** | 0.515 | 0.815 | **0.300** | 0.647 | 0.468 | **0.491** | 0.526 | 0.815 | 0.250 | 0.706 | **0.500** |

to train an LLM $f_\theta$ based on supervised fine-tuning:

$$\theta^* = \arg\min_\theta \frac{1}{n} \sum_{i=1}^{n} \mathcal{L}\left(f_\theta(q_i), R_i\right) \tag{5}$$

Once $\theta^*$ is obtained, the fine-tuned $f_{\theta^*}$ can generate responses for different queries.

In the fusion-stage matrix, model-level fusion represents late-stage parametric consolidation. Compared with query-level and thought-level fusion, it requires access to a trainable base model and introduces offline fine-tuning cost. However, once training is completed, inference only requires a single fused model, which can be desirable when users need deployment simplicity or cannot call multiple LLMs during inference. Studying this stage together with query- and thought-level fusion allows us to compare whether multi-LLM logs are more effectively used as routing supervision, prompting guidance, or training data.

## 6.2 Experimental Settings: Model-level Fusion

**Model and Implementation**. To evaluate model-level fusion, we fine-tune the best-performing small LLM (LLaMA-3.1 (8B)) on `LLMFusionBench`. For each query, we rank and select LLM-generated responses using three strategies from Section 2.1: (1) **Task Performance**: top-$K$ by performance; (2) **LLM Judge**: top-$K$ by judge score; (3) **Hybrid**: first by performance, then tie-break with judge score. The selected responses from each strategy serve as training data.

Following knowledge distillation practices (Hinton et al., 2015; Xie et al., 2020), we augment training set by including each query's ground truth along with the top-$K$ LLM responses. This forms the **Top-$K$ SFT** method. We compare it with two baselines: **Zero-Shot:** directly uses LLaMA-3.1 (8B) without fine-tuning; **Top-$K$ Label-only SFT:** trains only on repeated ground truths (replicated $K+1$ times to match training data volume). This setup tests whether model-level fusion effectively distills multi-LLM knowledge beyond supervised fine-tuning on ground truth alone.

## 6.3 Result Analysis: Model-level Fusion

We summarize results in Table 5 with three key observations: **(1) Model-level fusion yields modest gains that are consistent across our reported configurations.** Computing the cross-domain mean of Table 5 for each Top-$K$ SFT configuration against the zero-shot baseline of 0.530, every Top-$K$ SFT configuration improves the cross-domain mean by +0.012 to +0.024 (range: Performance-based +0.012 at $K$=10 to LLM-Judge +0.024 at $K$=5). Per-domain strict-win counts vary more: counting wins across all six Top-$K$ SFT configurations, model-level fusion beats zero-shot on **Commonsense (6/6)**, **Popular (6/6)**, **Math (5/6)**, and **Reading (4/6)**; the two domains where it usually loses are **Code (2/6)** and **World Knowledge (1/6, with 2 additional ties at 0.300)**. Top-$K$ Label-only SFT (rows 2 and 4 of Table 5) is reported as an ablation isolating the contribution of multi-LLM responses: at $K$=5, multi-LLM Top-$K$ SFT outperforms Label-only by +0.102 on the cross-domain mean (0.549 vs 0.447); at $K$=10, by +0.05 (0.546–0.548 vs 0.496). The gap is largest on World Knowledge, where Label-only collapses to 0.150 at both $K$ values, below the zero-shot baseline of 0.300. All model-level results are reported under a single SFT seed and a single response-sample draw; multi-seed variance estimation is left to future work. **(2) Fusion may**

Table 6: **Comparison across three fusion levels and six domains.** Commonsense, World, and Read denote Commonsense Reasoning, World Knowledge, and Reading Comprehension. Average indicates mean task performance. **Bold** and underline highlight the best and second-best.

| Fusion Level | Fusion LLM Size | Fusion Criteria | Math | Code | Commonsense | World | Read | Popular | Average |
|---|---|---|---|---|---|---|---|---|---|
| Best LLM | Full | N/A | **0.612** | 0.545 | 0.786 | **0.345** | 0.872 | 0.441 | 0.600 |
| Zero-shot | Small | N/A | 0.602 | 0.517 | 0.715 | 0.334 | 0.834 | 0.398 | 0.567 |
| Self-Consistency | Full | Response-level | 0.610 | 0.535 | 0.758 | 0.326 | 0.884 | 0.420 | 0.589 |
| LLM-Blender | Full | Response-level | 0.604 | 0.548 | 0.775 | 0.332 | 0.918 | 0.438 | 0.603 |
| FuseLLM-adapted | Full | Model-level | 0.542 | 0.455 | 0.792 | 0.286 | 0.846 | 0.418 | 0.557 |
| Query | Small | Performance | 0.599 | 0.522 | 0.792 | 0.338 | 0.862 | 0.395 | 0.585 |
| Thought | Small | Hybrid-8b | 0.610 | 0.423 | 0.778 | 0.301 | 0.956 | 0.443 | 0.585 |
| Thought | Full | Hybrid-8b | 0.544 | **0.564** | 0.799 | 0.304 | 0.965 | 0.452 | 0.605 |
| Thought | Small | Hybrid-70b | **0.612** | 0.550 | 0.778 | 0.312 | 0.966 | 0.447 | 0.611 |
| Thought | Full | Hybrid-70b | 0.588 | 0.562 | 0.796 | 0.303 | **0.970** | **0.473** | **0.615** |
| Thought | Full | Hybrid-70b w/o near-dup | 0.576 | 0.560 | 0.774 | 0.297 | 0.958 | 0.461 | 0.604 |
| Model | Small | LLM Judge | 0.530 | 0.510 | 0.708 | 0.286 | 0.851 | 0.410 | 0.549 |
| Model | Full | LLM Judge | 0.528 | 0.436 | **0.802** | 0.290 | 0.851 | 0.425 | 0.555 |

**hurt on code; we conjecture data scarcity and task mismatch as contributing factors.** All fusion strategies perform worse on code. We conjecture that data scarcity (MBPP and HumanEval together provide the smallest training pool of any domain — 494 samples / 6.97M tokens vs $\geq 900$ samples / $\geq 8.5$M tokens for every other domain, Table 10) and the divergence between code generation and QA-style supervision contribute to this result; verifying this hypothesis is left to future work. **(3) On the cross-domain mean, LLM-Judge or Hybrid selection dominates metric-based selection.** Computing the cross-domain mean of Table 5, LLM-Judge selection achieves the highest mean at Top-5 SFT (0.554, driven primarily by Code), and Hybrid selection is best at Top-10 SFT (0.548); Performance-based selection is dominated by at least one alternative at both $K$ values. We therefore recommend LLM-Judge or Hybrid selection over rigid metric-based selection, particularly in domains such as Code where exact-match metrics undervalue informative responses.

## 7 Comparisons Across Three `FusionFactory` Fusion Levels

In this section, we compare the three `FusionFactory` levels on the full test set of 14 tasks (Table 6), selecting the best-performing configuration for each level: Query-level: GraphRouter using task performance as the routing criterion; Thought-level: Hybrid strategy with thought summarization; Model-level: Fine-tuning with top-$K$ responses selected by the LLM Judge. For the original cross-level comparison, we use a unified generation setting to compare query-, thought-, and model-level fusion. In the revised table, we further add the strongest individual LLM among all 20 models, representative response/model-level fusion baselines, and a near-duplicate-controlled thought-level setting to provide a more complete comparison. Table 6 further breaks down fusion settings by two factors: Fusion LLM Size (whether responses are fused from all or only small models) and Fusion Criteria (how responses are selected, *e.g.*, task performance vs. LLMScore).

This cross-level comparison is central to the motivation of our fusion-stage matrix. Existing routing and distillation methods often study a single stage of the LLM pipeline in isolation, such as query-level model selection or model-level capability transfer. In contrast, our comparison asks how the same structured multi-LLM log data should be used across different intervention points: before generation for routing, during generation for thought-level reasoning augmentation, or after response collection for supervised fine-tuning. Therefore, Table 6 is not merely an engineering comparison, but a decision-oriented matrix for understanding the performance, cost, and generalization trade-offs among different ways of exploiting multi-LLM logs.

Consistent with the empirical nature of this work, our goal here is not to derive formal theoretical guarantees or closed-form complexity bounds. Instead, we use the unified benchmark setting to provide an empirical trade-off analysis of the three fusion stages. Query-level fusion tests whether logs are best used as routing supervision, thought-level fusion tests whether logs are best used as reusable reasoning guidance, and model-

level fusion tests whether logs are best used as distillation data. This comparison helps characterize the practical behavior of different fusion stages under the same data source and evaluation protocol.

Using unified experimental settings, we now summarize and compare the results across all three fusion levels. **(1) Thought-level fusion achieves the best overall performance.** Thought-Full-Hybrid-70B outperforms all settings, demonstrating its ability to integrate reasoning from diverse LLMs. However, performance gains plateau when increasing model size or response count, suggesting diminishing returns relative to cost. After adding the strongest individual LLM and representative fusion baselines, Thought-Full-Hybrid-70B still achieves the best average performance, outperforming Best LLM (0.615 vs. 0.600), Self-Consistency (0.615 vs. 0.589), LLM-Blender (0.615 vs. 0.603), and FuseLLM-adapted (0.615 vs. 0.557). This indicates that abstract thought-template retrieval provides benefits beyond selecting the strongest single model or aggregating final responses. This result suggests that multi-LLM logs are especially useful when their high-quality reasoning traces are reused as non-parametric guidance at inference time. Compared with pure routing, thought-level fusion can expose the responding model to diverse reasoning patterns rather than selecting only one model. Compared with model-level fusion, it avoids compressing heterogeneous task behaviors into a single set of parameters. Empirically, this gives thought-level fusion a favorable balance between flexibility and effectiveness, especially for tasks where reusable reasoning patterns are more valuable than simply selecting a model. The near-duplicate-controlled setting further supports this interpretation. After removing retrieved training queries with embedding similarity above 0.9, thought-level fusion still achieves 0.604 average performance, remaining above zero-shot (0.567) and close to the original thought-level result (0.615). This suggests that the gain is not solely driven by retrieving near-duplicate training queries, but also reflects reusable reasoning patterns captured by the thought templates.

**(2) Model-level fusion underperforms despite access to high-quality responses.** Fine-tuned models outperform zero-shot in only 3 of 6 domains, and even fall behind on average. We hypothesize two contributing factors — overfitting to multi-response inputs and difficulty abstracting across heterogeneous task styles during SFT — and view rigorous verification (e.g., per-task SFT ablations) as future work. This observation highlights an important limitation of treating multi-LLM logs purely as distillation data. Although model-level fusion has deployment advantages after training, the offline consolidation process may introduce training bias, reduce adaptability, or overfit to the particular response styles selected from the logs. In an empirical bias-generalization interpretation, model-level fusion can reduce inference-time variability by using a single fine-tuned model, but it may increase training-induced bias when heterogeneous responses from different tasks and LLMs are compressed into one base model. This explains why API-compatible distillation alone does not fully address the broader question of how multi-LLM logs should be used. The adapted FuseLLM baseline shows a similar trend, achieving 0.557 average performance, which is close to our model-level fusion result but still below thought-level fusion. This further suggests that directly compressing heterogeneous multi-LLM logs into model parameters may be less effective than using them as retrieval-time reasoning guidance.

**(3) Query-level fusion achieves moderate accuracy with the lowest cost.** Query-level fusion via GraphRouter avoids running the backbone on most queries and therefore achieves the lowest per-query cost: under cost-aware routing, GraphRouter cost is 0.0184 vs 0.1850 for Best-LLM under Performance First, a roughly 10× reduction (Table 4). Thought- and model-level fusion in this section's unified setting incur full LLaMA-3 (8B) inference cost per query, computed as the LLaMA-3 (8B) per-token price (Table 8) applied to the per-query token counts available in `LLMFusionBench` (Table 9). Thought-level fusion additionally incurs ∼5% latency overhead and 0.03s retrieval overhead over zero-shot (Appendix D, Table 11). The cost ordering across fusion levels follows directly from this routing-vs-full-inference distinction. This result confirms the value of query-level fusion for cost-sensitive API-serving scenarios. Since query-level fusion only requires a lightweight router and calls one selected LLM at inference time, it provides a practical option when users prioritize efficiency. However, its performance ceiling can be limited because it exploits multi-LLM logs only through model selection, without reusing richer reasoning traces or consolidating capabilities into parameters. Empirically, query-level fusion therefore has a low-overhead profile, but may be biased toward the capability boundary of the selected LLM. Compared with response-level baselines, query-level fusion achieves lower average performance than LLM-Blender (0.585 vs. 0.603), but it requires only one selected

model call rather than aggregating multiple candidate responses. This highlights its distinct role as an efficiency-oriented fusion strategy rather than a purely performance-oriented one.

**(4) Fusion provides only marginal improvements on World Knowledge and Math.** On these domains, the best fusion result exceeds zero-shot by at most 2 points absolute (Table 6: Math thought-level 0.612 vs zero-shot 0.602; World Knowledge query-level 0.338 vs zero-shot 0.334), suggesting these domains — which depend heavily on factual recall and strict logical sequencing — are less amenable to multi-LLM fusion under our current framework. These domain-level results further support the need for a fusion-stage matrix. Different tasks benefit from different ways of using multi-LLM logs. Knowledge-intensive tasks may require strict factual grounding and are sensitive to noisy fused information, while mathematical tasks require coherent step-by-step reasoning and may be harmed by inconsistent reasoning traces. Therefore, no single fusion level uniformly dominates across all settings, and a systematic comparison is needed to determine which fusion stage is appropriate under different domain requirements.

Taken together, these observations provide an empirical interpretation of the trade-offs among the three fusion levels. Query-level fusion is efficient and easy to deploy, but it mainly benefits from better model selection. Thought-level fusion is more flexible because it reuses reasoning traces without modifying model parameters, but it introduces retrieval and prompt-token overhead. Model-level fusion is attractive when a single deployable model is desired, but it requires offline training and may generalize poorly when the logs contain heterogeneous task patterns. The additional comparisons with Best LLM, response-level fusion baselines, and near-duplicate-controlled retrieval further show that the gains of thought-level fusion are not simply due to a stronger single model, final-response aggregation, or near-duplicate retrieval. This empirical trade-off analysis complements the stage-aware taxonomy in Section 3.

Overall, these findings validate our central claim: multi-LLM log data, when systematically structured by query / task / response triples and ranked by performance and LLM-judge scores, supports fusion strategies that outperform single-LLM baselines. On the full test set (Table 6), thought-level fusion improves over zero-shot by 4.8 points on average across the six domains ($0.567 \rightarrow 0.615$); under cost-aware routing, query-level fusion matches the strongest single LLM at an order of magnitude lower cost (Table 4: GraphRouter 0.602 reward / 0.0184 cost vs Best-LLM 0.546 / 0.1850 under Performance First). The framework supports both local and API-served LLMs and applies across all three fusion stages, and `LLMFusionBench` provides a demanding benchmark for real-world multi-LLM integration. More importantly, the comparison demonstrates the unique value of our work beyond existing API-only routing or distillation methods. The main contribution is not simply that a method can operate with API-accessible models, but that the same multi-LLM log source can be systematically evaluated across query-level, thought-level, and model-level fusion. This matrix reveals when logs are better used as routing supervision, when they are better used as reasoning templates, and when they are less effective as fine-tuning data. Such cross-stage evidence would be difficult to obtain from studies that focus only on routing, response-level fusion, or distillation.

## 8 Additional Related Work

**LLM fusion**. Recent work explores fusing multiple LLMs or reasoning traces to enhance performance, robustness, and generalization. Query-level methods (*e.g.*, LLMRouter (Li et al., 2023c), STaR (Zhou et al., 2022)) treat routing as policy learning to select the best model per query. Thought-level methods (*e.g.*, Thought Propagation (Li et al., 2023d), Tree of Thoughts (Yao et al., 2023)) aggregate reasoning paths for more consistent answers. Model-level methods (*e.g.*, FrugalGPT (Jiang et al., 2023c), Self-Consistency (Wang et al., 2022)) combine multiple model outputs or samples to approximate stronger reasoning while controlling cost. LLM-as-judge approaches (*e.g.*, LLM-as-Judge (Zhang et al., 2023), RAG Fusion (Lewis et al., 2021)) evaluate and select responses via a secondary LLM. However, most prior studies focus on a single level of fusion. We instead introduce `LLMFusionBench`, a large-scale benchmark for LLM fusion, and conduct a systematic comparison across three levels via `FusionFactory`, enabling more holistic and integrated insights.

**LLM Router**. Recent work on LLM routing focuses on improving efficiency and response quality by directing queries to the most suitable model. Early methods like RouteLLM (Ding et al., 2024b) optimize preference-aligned model selection via learned policies, while RouterBench (Hu et al., 2024) benchmarks

routing decisions but does not study routing logs for capability fusion. TO-Router (Stripelis et al., 2024) and Expert Router (Anonymous, 2024) emphasize parallel dispatch for efficiency, and Glider (Li et al., 2024a) integrates global and local signals to refine expert selection. In contrast, our work focuses on capability-oriented analysis. We leverage large-scale multi-LLM log data not only for model selection, but as a form of weak supervision to systematically study and fuse LLM capabilities across three flexible levels.

# 9 Conclusion

In this work, we revisit LLM fusion through the lens of API-based multi-LLM log data, arguing that practical fusion must (1) be compatible with real-world serving (local and API) and (2) support integration at different pipeline stages. We introduce `LLMFusionBench`, a large-scale benchmark spanning 14 tasks and 20 LLMs with diverse response modes and reusable thought templates, and propose `FusionFactory`, a three-level framework for stage-aware fusion. Experiments show that thought-level fusion delivers the strongest gains, query-level fusion offers the best accuracy–cost trade-off, and model-level fusion lags due to generalization limits. Overall, these results validate multi-LLM log data as a practical supervision source and position `LLMFusionBench` as a demanding, real-world testbed for flexible, serving-compatible LLM fusion methods. `FusionFactory` is model-agnostic by construction: extending the LLM pool to newer frontier models (e.g., GPT-5, Claude 4, Gemini 2) requires only re-running response collection on the existing query set, with no architectural change to the fusion pipeline.

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

# A Limitations and Future Work

While our work focuses on evaluating real-world task performance - particularly in domains like math and coding - future research should also investigate aspects of trustworthiness (Qi et al., 2024; Han et al., 2024b) and justice (Han et al., 2024a; Dai et al., 2024a) in the fusion process. The act of fusing multiple responses may inadvertently introduce social biases, inaccuracies, or conflicting reasoning, which could impact the reliability and ethical soundness of the LLMs. Exploring safeguards or calibration techniques to ensure more robust and equitable fusion outcomes presents a valuable direction for further study.

Our study focuses on a practical and broadly accessible form of multi-LLM log data that can be collected in both local and API-based serving scenarios. Specifically, `LLMFusionBench` records aligned responses from multiple LLMs for the same query, together with model identity, response mode, inference cost, task-specific performance, and LLM-judge scores. These fields provide useful supervision signals for query-level routing, thought-level template construction, and model-level distillation. This design choice is intentional because, in many real-world deployments, especially when using closed-source or API-based LLMs, users may only have access to textual responses and serving metadata rather than model logits, internal states, or training-time information.

Richer production logs may additionally contain interaction-level signals, such as temporal order, explicit user feedback, error patterns, user revision behavior, and multi-turn interaction history. These signals can potentially enable more fine-grained fusion strategies. For example, temporal order may help track how model behavior changes over time, user feedback may provide direct preference supervision, and interaction history may support personalized or context-aware fusion. However, such signals are not always consistently available across public benchmarks, local models, and API-based serving systems. Incorporating these richer interaction-level log signals is therefore an important direction for future extensions of `LLMFusionBench` and `FusionFactory`.

A related concern is the privacy risk associated with log-based fusion. Multi-LLM logs may contain user queries, model responses, feedback, and interaction traces, which can include sensitive or personally identifiable information in real-world deployments. Although `LLMFusionBench` is constructed from public evaluation datasets and does not involve private user data, practical systems built on multi-LLM logs should adopt privacy-aware data handling procedures. These may include anonymization, personally identifiable information filtering, access control, retention policies, and aggregation or synthetic-log generation before logs are used for retrieval, training, or distillation.

Another limitation is the potential negative effect of low-quality logs. Since `FusionFactory` relies on logged responses and feedback signals, noisy, biased, hallucinated, or incorrectly scored responses may propagate into the fusion process. In query-level fusion, low-quality logs may lead to suboptimal routing policies. In thought-level fusion, misleading responses may be summarized into unhelpful thought templates. In model-level fusion, noisy or biased responses may be amplified through supervised fine-tuning. Future work should therefore investigate stronger log-quality control mechanisms, such as uncertainty-aware filtering, consistency checking, human or model-assisted verification, and robust training objectives that reduce the influence of unreliable logged responses.

In addition to log quality, API-based deployment of fusion systems may introduce reliability and transparency risks. A query-level router may select an unsuitable model, and a thought-level retriever may retrieve irrelevant or misleading templates, causing errors that are difficult to diagnose from the final response alone. Practical fusion systems should therefore expose routing decisions, retrieved templates, confidence estimates, and fallback mechanisms whenever possible, so that users and developers can better audit and control the fusion process.

Our current benchmark mainly focuses on single-turn benchmark queries, while real-world LLM logs often involve long documents, evolving user intents, tool-use traces, and multi-turn interaction histories. These settings can introduce additional failure modes. For long-text inputs, retrieved thought templates may omit important context or conflict with document-specific evidence. For multi-turn dialogue, query-level similarity alone may fail to capture changing user goals, and model-level distillation may compress incomplete or stale interaction states. Extending `LLMFusionBench` and `FusionFactory` to long-context reasoning, multi-

turn dialogue, tool-augmented agents, and distribution-shifted deployment settings is an important future direction.

This work is primarily an empirical benchmark and framework study. We focus on constructing a large-scale multi-LLM log benchmark and systematically evaluating how the same log source can be used across query-level, thought-level, and model-level fusion. Therefore, our goal is not to provide formal theoretical guarantees or closed-form complexity bounds. A more formal theoretical treatment of multi-LLM fusion remains an important future direction. For example, future work could develop formal criteria for selecting fusion stages based on query difficulty, uncertainty, domain shift, cost constraints, or the expected reliability of retrieved logs.

Although several prior routing and distillation methods already support API-only settings, our study emphasizes a broader fusion-stage matrix rather than API compatibility alone. Existing routing methods mainly exploit logs for query-level model selection, while distillation methods mainly use model outputs for model-level capability transfer. In contrast, `FusionFactory` studies how the same structured multi-LLM log data can support routing, reasoning-template augmentation, and supervised fine-tuning under a unified benchmark and evaluation protocol. This matrix provides a practical decision framework for choosing how to use multi-LLM logs under different deployment constraints.

Future work could further extend this fusion-stage matrix by developing hybrid or adaptive fusion strategies. For example, a system may use query-level fusion for simple or cost-sensitive queries, invoke thought-level fusion for reasoning-intensive queries, and apply model-level fusion when repeated high-quality logs are available in a stable domain. Such adaptive strategies could dynamically choose the appropriate fusion level according to task difficulty, budget, domain type, or uncertainty estimates, further improving the utility of multi-LLM log data in real-world deployments.

## B More on `LLMFusionBench`

Here, we list all the tasks and the metrics used for each in Table 7, and all models along with their sizes and costs in Table 8.

## C Router Baseline Description for Query-Level Fusion

For the query-level fusion, we evaluate five representative routing methods to assess their LLM fusion capabilities: (1) *RouterKNN* (Shnitzer et al., 2023), a non-parametric baseline that routes by nearest neighbors in query space and selects the majority LLM label; (2) *RouterSVM* (Hu et al., 2024), a support vector machine trained on query features and task labels; (3) *RouterMLP* (Shnitzer et al., 2023), a multi-layer perceptron leveraging query embeddings and task context; (4) *RouterBERT* (Ong et al., 2024), a compact BERT classifier that encodes both query and task to predict the optimal LLM; and (5) *GraphRouter* (Feng et al., 2024b), a graph-based model that treats routing as node classification over a heterogeneous graph of queries, tasks, and LLMs with learned edge interactions. All models use `all-MiniLM-L6-v2` for embeddings, except GraphRouter, which employs Longformer (Beltagy et al., 2020) to obtain the embeddings for task/query/LLM. We also include several static baselines: *Best LLM*, which achieves the highest performance on the training set; *LLM-Mean*, which yields the average test performance across all LLMs; and *Largest LLM*, which denotes the test performance of the largest model.

## D Inference Time for Thought-Level Fusion

**Setup and offline preprocessing.** As introduced in Table 1 and Figure 1, the thought templates used by the thought-level fusion style are summarized *offline* on `LLMFusionBench`. For new queries, we do not regenerate summaries; we only retrieve the most relevant thought templates and prepend them to the prompt. Retrieval is implemented with FAISS, and responses are produced using the NVIDIA NIM API. This design minimizes online overhead and keeps inference practical for API-based serving.

Table 7: **The tasks and corresponding evaluation metrics of the dataset used in constructing `LLMFusionBench`, organized by domain.**

| Domain | Tasks | Metrics |
|---|---|---|
| Math | GSM8K (Cobbe et al., 2021) | Accuracy |
| | MATH (Hendrycks et al., 2021b) | Accuracy |
| Code | MBPP (Austin et al., 2021) | Pass@1 |
| | HumanEval (Chen et al., 2021) | Pass@1 |
| Commonsense Reasoning | CommonsenseQA (Talmor et al., 2019) | Accuracy |
| | ARC (Clark et al., 2018) | Accuracy |
| | HellaSwag (Zellers et al., 2019) | Accuracy |
| | OpenBookQA (Mihaylov et al., 2018) | Accuracy |
| World Knowledge | NaturalQuestions (Kwiatkowski et al., 2019) | CEM |
| | TriviaQA (Joshi et al., 2017) | CEM |
| Reading Comprehension | SQuAD (Rajpurkar et al., 2016) | CEM |
| | BoolQ (Clark et al., 2019) | CEM |
| Popular | MMLU (Hendrycks et al., 2021a) | Multi-task Accuracy |
| | GPQA (Rein et al., 2023) | Accuracy |

Table 8: **Language Models and estimated price (in $ per 1M tokens).**

| Size Type | Model | Size | Input Price | Output Price |
|---|---|---|---|---|
| Small | Gemma-2 (9b) (Team et al., 2024b) | 9B | 0.20 | 0.20 |
| | Qwen2 (7b) (Yang et al., 2024a) | 7B | 0.20 | 0.20 |
| | Gemma (7b) (Team et al., 2024b) | 7B | 0.20 | 0.20 |
| | CodeGemma (7b) (Team et al., 2024a) | 7B | 0.20 | 0.20 |
| | Mistral (7b) (Jiang et al., 2023a) | 7B | 0.20 | 0.20 |
| | LLaMA-3.1 (8b) (Grattafiori et al., 2024) | 8B | 0.20 | 0.20 |
| | Granite (8b) (Mishra et al., 2024) | 8B | 0.20 | 0.20 |
| | LLaMA-3 ChatQA (8b) (Liu et al., 2024) | 8B | 0.20 | 0.20 |
| | Qwen2.5 (7b) (Qwen et al., 2025) | 7B | 0.20 | 0.20 |
| | Mistral-Nemo (12b) (Mistral AI, 2024) | 12B | 0.30 | 0.30 |
| Medium | Granite Code (34b) (Mishra et al., 2024) | 34B | 0.80 | 0.80 |
| | LLaMA-3.3 Nemotron Super (49b)(Wang et al., 2024) | 49B | 0.90 | 0.90 |
| | LLaMA-3.1 Nemotron (51b) (Wang et al., 2024) | 51B | 0.90 | 0.90 |
| | Mixtral (8x7b) (Jiang et al., 2024) | 56B (8×7B) | 0.60 | 0.60 |
| | LLaMA-3.1 (70b)(Grattafiori et al., 2024) | 70B | 0.90 | 0.90 |
| | LLaMA-3 ChatQA (70b) (Liu et al., 2024) | 70B | 0.90 | 0.90 |
| | LLaMA-3 (70b) (Grattafiori et al., 2024) | 70B | 0.90 | 0.90 |
| Large | Palmyra Creative (122b) (team, 2024) | 122B | 1.80 | 1.80 |
| | Mixtral (8x22b)(Jiang et al., 2024) | 176B (8×22B) | 1.20 | 1.20 |
| | DeepSeek-R1 (671b) (Guo et al., 2025) | 671B | 0.55 | 2.19 |

**Inference-time comparison.** Table 11 reports average inference times on the full test set with `LLaMA-3 (8B)` as the backbone across three paradigms: Zero-Shot, Few-Shot, and Thought-Hybrid. Thought-level fusion incurs only a minor increase relative to Zero-Shot and Few-Shot, which is attributable to the extra input tokens from the retrieved thought templates. The average lookup time for the thought-based method is **0.03 s**, accounting for **0.7%** of the total Thought-Hybrid inference time, which is essentially negligible.

Table 9: **Data collection information.**

| Collected Data Fields | Task Name; Task Description; Task Description Embedding; Query; Query Embedding; Ground Truth; Metric; LLM; Input Price; Output Price; Input Tokens Num; Output Tokens Num; Performance; Cost; Response; LLM Description |
|---|---|

Table 10: **Sample counts in the `LLMFusionBench` training set, partial test set, and full test set across different tasks.**

| Domain | Tasks | Train | Train Tokens | Partial Test | Test Tokens | Full Test |
|---|---|---|---|---|---|---|
| Math | GSM8K (Cobbe et al., 2021) | 500 | 5.838M | 50 | 551K | 1,319 |
| | MATH (Hendrycks et al., 2021b) | 500 | 8.591M | 50 | 944K | 5,000 |
| Code | MBPP (Austin et al., 2021) | 374 | 4.702M | 50 | 623K | 500 |
| | HumanEval (Chen et al., 2021) | 120 | 2.268M | 44 | 836K | 44 |
| Commonsense Reasoning | CommonsenseQA (Talmor et al., 2019) | 500 | 4.511M | 50 | 423K | 1,221 |
| | ARC (Clark et al., 2018) | 500 | 5.387M | 50 | 489K | 1,172 |
| | HellaSwag (Zellers et al., 2019) | 500 | 7.969M | 50 | 795K | 10,042 |
| | OpenBookQA (Mihaylov et al., 2018) | 500 | 4.496M | 50 | 431K | 500 |
| World Knowledge | NaturalQuestions (Kwiatkowski et al., 2019) | 500 | 4.792M | 50 | 444K | 3,610 |
| | TriviaQA (Joshi et al., 2017) | 500 | 3.722M | 50 | 323K | 17,944 |
| Reading Comprehension | SQuAD (Rajpurkar et al., 2016) | 500 | 6.572M | 50 | 646K | 10,570 |
| | BoolQ (Clark et al., 2019) | 500 | 7.101M | 50 | 671K | 3,270 |
| Popular | MMLU (Hendrycks et al., 2021a) | 500 | 10.515M | 50 | 628K | 14,042 |
| | GPQA (Rein et al., 2023) | 400 | 10.044M | 44 | 1.2M | 44 |

In return, the performance gains are substantial. We also expect the relative overhead to further diminish with larger base models, since total inference time increases while lookup time remains nearly constant.

Table 11: **Average inference time (seconds) on the full test set with `LLaMA-3` (8B) as the backbone.** Thought-level fusion adds minimal overhead due to template tokens, while retrieval adds only 0.03 s on average.

| Method | Mean Inference Time (s) |
|---|---|
| Zero-Shot | 3.74 |
| Few-Shot | 3.89 |
| Thought-Hybrid | 3.91 |

## E   Prompts for Thought-Level Fusion

Previous studies have shown that thought templates can enhance the performance of LLMs (Feng et al., 2024a; Yang et al., 2024c). However, existing templates are often either too general to a big category of questions (Yang et al., 2024c) or too specific, tailored to only one particular case (Feng et al., 2024a). Our prompt 12 is designed to strike a balance between these extremes: it is specific enough to guide similar problems effectively, yet general enough to be applicable across a range of variations.

## F   Detailed Results for Comparison Across Three `FusionFactory` Levels

In this section, we present detailed results for each task across all methods from the three aforementioned fusion levels, as shown in Table 14.

Table 12: **Prompts for thought template creation**.

---

Given this question and example solutions, extract a concise thought template that captures the effective reasoning pattern and can serve as guidance:

**Question:** [QUERY]

**Here are 3 high-performing solutions:**
Solution 1: [Response 1]
Solution 2: [Response 2]
Solution 3: [Response 3]

**Please create a concise and clear thought template (1–5 sentences total) focusing on:**

- **Core Task Summarization:** Identify the core problem type and general approach needed (1 sentence).

- **Reasoning Step:** Provide a clear chain of thought to address this problem (1–3 sentences).

- **Answer Template:** Describe the preferred answer format or structure (1 sentence).

**Your template should be specific enough to guide similar problems but general enough to work across variations.**

**Thought Template:**

---

Table 13: **The model names and their specific settings.** We introduce these model names in Table 14 in terms of the aspects corresponding to Table 6, namely Fusion Level, Fusion LLM Size, and Fusion Criteria.

| Model Name | Fusion Level | Fusion LLM Size | Fusion Criteria |
|---|---|---|---|
| **Zero-shot** | Zero-shot | Small | N/A |
| **Query-Small** | Query | Small | Performance |
| **Thought-Small-8b** | Thought | Small | Hybrid-8b |
| **Thought-Full-8b** | Thought | Full | Hybrid-8B |
| **Thought-Small-70b** | Thought | Small | Hybrid-70B |
| **Thought-Full-70b** | Thought | Full | Hybrid-70b |
| **Model-Small** | Model | Small | LLM Judge |
| **Model-Full** | Model | Full | LLM Judge |

## G   More Analysis: Does Reasoning Data Help for Reasoning-Oriented Tasks?

**Motivation.**   As discussed around thought-level fusion, `LLMFusionBench` includes both reasoning and non-reasoning responses (LLM-think vs. LLM-direct).

**Analysis setup.**   We analyze, under the *thought-level hybrid* selection setting, the composition of best-selected responses per task (reasoning vs. non-reasoning). This directly tests whether reasoning-style data is preferentially selected for tasks that demand more reasoning.

Table 14: **Detailed results for all levels of fusion across all benchmarks.**

| Scenario | Task | Zero-shot | Query-Small | Thought-Small-8b | Thought-Full-8b | Thought-Small-70b | Thought-Full-70b | Model-Small | Model-Full |
|---|---|---|---|---|---|---|---|---|---|
| Math | GSM8K | 0.749 | 0.7983 | 0.771 | 0.651 | 0.771 | 0.749 | 0.715 | 0.742 |
| | MATH | 0.455 | 0.4000 | 0.449 | 0.437 | 0.452 | 0.427 | 0.344 | 0.314 |
| Code | HumanEval | 0.432 | 0.4545 | 0.546 | 0.530 | 0.500 | 0.523 | 0.566 | 0.485 |
| | MBPP | 0.601 | 0.5900 | 0.300 | 0.598 | 0.600 | 0.600 | 0.455 | 0.386 |
| Commonsense Reasoning | ARC (Challenge) | 0.736 | 0.8276 | 0.819 | 0.839 | 0.819 | 0.835 | 0.689 | 0.812 |
| | CommonsenseQA | 0.723 | 0.7371 | 0.757 | 0.770 | 0.757 | 0.766 | 0.716 | 0.792 |
| | HellaSwag | 0.649 | 0.7424 | 0.745 | 0.757 | 0.745 | 0.750 | 0.727 | 0.794 |
| | OpenBookQA | 0.750 | 0.8620 | 0.792 | 0.830 | 0.792 | 0.834 | 0.700 | 0.810 |
| World Knowledge | NaturalQuestions | 0.473 | 0.4716 | 0.409 | 0.414 | 0.427 | 0.416 | 0.404 | 0.414 |
| | TriviaQA | 0.195 | 0.2034 | 0.192 | 0.193 | 0.197 | 0.189 | 0.168 | 0.165 |
| Reading Comprehension | SQuAD | 0.825 | 0.8746 | 0.921 | 0.947 | 0.948 | 0.950 | 0.849 | 0.863 |
| | BoolQ | 0.843 | 0.8492 | 0.991 | 0.983 | 0.985 | 0.990 | 0.852 | 0.838 |
| Popular | GPQA | 0.186 | 0.1818 | 0.273 | 0.227 | 0.273 | 0.273 | 0.205 | 0.182 |
| | MMLU | 0.610 | 0.6073 | 0.612 | 0.676 | 0.621 | 0.672 | 0.614 | 0.668 |

**Findings.** Table 15 shows that (1) reasoning ratios are higher for most reasoning-oriented tasks—**Math**, **Code**, **Commonsense Reasoning**, and **Popular**—indicating that such tasks benefit from reasoning traces. (2) In contrast, **World Knowledge** and **Reading Comprehension** skew toward non-reasoning outputs, consistent with their stronger reliance on factual recall and concise evidence extraction. Overall, these results support that reasoning-level information in `LLMFusionBench` aids selection for reasoning-heavy tasks while preserving flexibility for knowledge-intensive ones.

Table 15: **Reasoning vs. Non-Reasoning ratios under thought-level hybrid selection (by domain).**

| Task Domain | Reasoning Ratio (%) | Non-Reasoning Ratio (%) |
|---|---|---|
| Math | 51.25 | 48.75 |
| Code | 53.18 | 46.82 |
| Commonsense Reasoning | 71.00 | 29.00 |
| World Knowledge | 37.90 | 62.10 |
| Reading Comprehension | 44.10 | 55.90 |
| Popular | 57.50 | 42.50 |

## H Implementation details of `FusionFactory`

We utilize NVIDIA API for our API calling in data generation and model inference. We also use a NVIDIA A6000 GPU to obtain all the embeddings for queries/tasks/LLMs described in section 4.2.

For model-level fusion, we adopt LLaMA-Factory to fine-tune the model with LoRA, which is conducted on NVIDIA A6000 GPUs. For LoRA, we apply adaptation to all transformer layers with a rank of 8. Inputs are processed with a maximum context length of 2048 tokens unless otherwise specified. The training uses a per-device batch size of 8 and a gradient accumulation step of 4, resulting in an effective batch size of 32. We set the learning rate to 1e-4 and train the model for 3 full epochs. A cosine learning rate scheduler is used with a warmup ratio of 0.1 to ensure stable convergence. Training is conducted using bfloat16 precision, and Flash Attention 2 is enabled to accelerate attention computation. Note that, as shown in Table 10, the training data in the Code domain is relatively smaller and more challenging compared to other domains. Therefore, following the approach adopted in previous work (Roziere et al., 2023; Li et al., 2023a), we train a separate model for the Code domain, while a unified model is trained for the remaining domains.

---

`https://build.nvidia.com/nim`
`https://github.com/hiyouga/LLaMA-Factory`

## I  Prompts for LLM-As-Judge Score Generation

We present the prompt for LLM-As-Judge Score Generation in Table 16.

Table 16: **Prompts for LLM-As-Judge score generation**.

---

You are an expert judge evaluating the quality of an AI model's response. Please score the response based on the following criteria:

1. **Correctness (0–1):** Is the answer correct according to the ground truth?

2. **Thought Process (0–1):** Does the response show clear reasoning and explanation?

3. **Training Data Quality (0–1):** Is the response well-structured and suitable for supervised fine-tuning?

**Query:** {query}
**Ground Truth:** {ground_truth}
**Response:** {response}

Please provide a single score from 0 to 1, where:

- 0: Incorrect answer

- 0.3: Correct answer but minimal thought process

- 0.6: Correct answer with some thought process

- 0.8: Correct answer with good thought process

- 1.0: Correct answer with excellent thought process and well-suited for training

Return the score in the following format:

`<answer>SCORE</answer>`

---

## J  Descriptions for tasks and LLMs

Using task and model descriptions generated by LLMs enhances the expressiveness and generalization capability of the router. In this work, we present descriptions of various tasks and LLMs generated by GPT-4o. Specifically, GPT-4o captures the unique characteristics and challenges associated with each task, as well as the size, cost, and distinct strengths of different LLMs. These detailed descriptions are provided in the tables below.

## K  Case Study for All Levels

To better understand the behavior of LLMs under different fusion strategies, we conduct a series of case studies spanning diverse task types, including Math, Code, Commonsense Reasoning, World Knowledge, Reading Comprehension, and Popular questions. Each example is evaluated under four levels: Zero-shot, Query-level, Thought-level, and Model-level. These comparative analyses provide qualitative insights into how reasoning granularity and intervention level impact model performance, faithfulness, and error patterns. See Tables 18, 19, 20, 21, 22, and 23 for detailed comparisons across tasks.

Table 17: **Comparison of performance and LLM judge for responses.** We selected responses from two different LLMs on the Math and GSM8K tasks, and obtained their respective Performance and LLM Score. It can be observed that while Performance effectively reflects the correctness of responses, it fails to distinguish the amount of information provided in responses that are all correct. In contrast, LLM Score compensates for this limitation by capturing the informational richness of responses, thereby offering greater potential for enhancing LLM capabilities fusion.

| Question of Math | |
|---|---|
| Simplify $(3p^3 - 5p + 6) + (4 - 6p^2 + 2p)$. Express your answer in the form $Ap^3 + Bp^2 + Cp + D$, where $A$, $B$, $C$, and $D$ are numbers (possibly negative). | |
| **Answer of Gemma-2 (9b)** | **Answer of Gemma-2-think (9b)** |
| $\boxed{3p^3 - 6p^2 - 3p + 10}$ | Here's how to simplify the expression: **1. Remove the parentheses:** Since we are adding, the parentheses don't affect the order of operations. $3p^3 - 5p + 6 + 4 - 6p^2 + 2p$ **2. Combine like terms:** Identify terms with the same variable and exponent, and combine their coefficients. $3p^3 - 6p^2 + (-5p + 2p) + (6 + 4)$ **3. Simplify:** $3p^3 - 6p^2 - 3p + 10$ $\boxed{3p^3 - 6p^2 - 3p + 10}$ |
| **Performance**: 1.0 **LLMScore**: 0.3 *Correct answer but minimal thought process* | **Performance**: 1.0 **LLMScore**: 0.8 *Correct answer with good thought process* |
| **Question of GSM8K** | |
| Mr. Maximilian has a rental building that he collects rent from every month. The number of units in the building is 100. If the building is 3/4 occupied for a whole year, and each resident of the building pays a rent of \$400, calculate the amount of money Mr. Maximilian receives in that year. | |
| **Answer of Qwen2.5 (7b)** | **Answer of Qwen2.5-think (7b)** |
| To calculate the amount of money Mr. Maximilian receives in a year, we can follow these steps: **1. Calculate the number of occupied units:** - The building has 100 units. - It is 3/4 occupied. - Number of occupied units $= (100 \times \frac{3}{4} = 75)$. **2. Calculate the monthly rent collected:** - Each resident pays \$400 per month. - Monthly rent collected $= (75 \times 400 = 30,000)$ dollars. **3. Calculate the annual rent collected:** - There are 12 months in a year. - Annual rent collected $= (30,000 \times 12 = 360,000)$ dollars. Therefore, Mr. Maximilian receives \$360,000 in rent for the year. | Sure, let's break down the problem step by step to find out how much money Mr. Maximilian receives in a year from the rent. **Step 1: Calculate the Number of Occupied Units** 1. The total number of units in the building is 100. 2. The building is 3/4 (or 75%) occupied. To find the number of occupied units: Number of occupied units $= \frac{3}{4} \times 100 = 75$ **Step 2: Calculate the Monthly Rent Collection** 1. Each resident pays a rent of \$400 per month. 2. There are 75 occupied units. To find the total monthly rent collection: Monthly rent collection $= 75 \times 400 = 30,000$ **Step 3: Calculate the Annual Rent Collection** 1. The monthly rent collection is \$30,000. 2. There are 12 months in a year. To find the total annual rent collection: Annual rent collection $= 30,000 \times 12 = 360,000$ **Final Answer** Mr. Maximilian receives \$360,000 in rent for the year. |
| **Performance Score**: 1.0 **LLMScore**: 0.9 *Correct answer with very good thought process* | **Performance Score**: 1.0 **LLMScore**: 1.0 *Correct answer with excellent thought process* |

Table 18: **Case study - Math.** An example math question used to compare model responses across different settings. This case probes multi-step numerical reasoning, arithmetic consistency, and symbolic manipulation under zero-shot, query-level, thought-level, and model-level settings.

| Question of Math | | | |
|---|---|---|---|
| Janet's ducks lay 16 eggs per day. She eats three for breakfast every morning and bakes muffins for her friends every day with four. She sells the remainder at the farmers' market daily for $2 per fresh duck egg. How much in dollars does she make every day at the farmers' market? | | | |
| **Zero-shot** | **Query-level** | **Thought-level** | **Model-level** |
| To find out how much Janet makes every day at the farmers' market, we need to first determine how many eggs she has left after eating and baking. Janet's ducks lay 16 eggs per day. She eats 3 for breakfast and bakes 4 for muffins, so she uses $3 + 4 = 7$ eggs. To find out how many eggs she has left, we subtract the number of eggs she uses from the total number of eggs laid: 16 - 7 $= 9$ eggs. Janet sells the remaining 9 eggs at the farmers' market for $2 per egg. To find out how much she makes, we multiply the number of eggs she sells by the price per egg: 9 x $2 = $18. Therefore, Janet makes $18 every day at the farmers' market. | To find out how much Janet makes every day at the farmers' market, we need to follow these steps: 1. Calculate the total number of eggs laid by the ducks per day. 2. Calculate the number of eggs Janet eats for breakfast and bakes for her friends. 3. Calculate the number of eggs left over after Janet eats and bakes. 4. Calculate the amount of money Janet makes from selling the leftover eggs. Step 1: Calculate the total number of eggs laid by the ducks per day. Janet's ducks lay 16 eggs per day. Step 2: Calculate the number of eggs Janet eats for breakfast and bakes for her friends. Janet eats 3 eggs for breakfast and bakes 4 eggs for her friends, so she uses a total of $3 + 4 = 7$ eggs. Step 3: Calculate the number of eggs left over after Janet eats and bakes. The total number of eggs laid is 16, and Janet uses 7 eggs, so the number of eggs left over is 16 - 7 = 9. Step 4: Calculate the amount of money Janet makes from selling the leftover eggs. Janet sells the leftover eggs for $2 each, and she has 9 eggs left over, so she makes 9 x $2 = $18. Therefore, Janet makes $18 every day at the farmers' market. `<answer>$18</answer>` | To solve this problem, I will use the thought template for calculating the remaining quantity of items after filling containers with a specific capacity. **Core Task Summarization**: Determine the daily earnings from selling eggs at the farmers' market. **Reasoning Step**: 1. Calculate the total number of eggs laid by the ducks per day: 16 eggs/day 2. Calculate the total number of eggs eaten by Janet for breakfast: 3 eggs/day 3. Calculate the total number of eggs used for baking muffins: 4 eggs/day 4. Calculate the total number of eggs used for breakfast and baking: $3 + 4 = 7$ eggs/day 5. Calculate the remaining number of eggs to be sold at the farmers' market: 16 - 7 $= 9$ eggs/day 6. Calculate the daily earnings from selling eggs at the farmers' market: 9 eggs/day * $2/egg $= $18/day **Answer Template**: The final answer is a numerical value representing the daily earnings from selling eggs at the farmers' market. The final answer is: $18 | To calculate how much Janet makes every day at the farmers' market, we need to determine how many eggs she has left after using some for breakfast and baking muffins. Then, we can calculate the remaining eggs and multiply that by the price per egg. 1. **Calculate the total eggs laid per day:** Janet's ducks lay 16 eggs per day. 2. **Calculate the eggs used for breakfast:** She eats three eggs for breakfast every morning. 3. **Calculate the eggs used for baking muffins:** She bakes muffins with four eggs every day. 4. **Calculate the total eggs used:** 3 (breakfast) + 4 (muffins) = 7 eggs. 5. **Calculate the remaining eggs:** 16 (total eggs) - 7 (eggs used) = 9 eggs. 6. **Calculate the money made at the farmers' market:** 9 eggs * $2 per egg = $18. So, Janet makes $18 every day at the farmers' market. |

Table 19: **Case study - Code.** A representative code generation problem requiring functional understanding and list manipulation in Python. The table shows how different levels affect the model's ability to parse task requirements, implement correct logic, and produce syntactically valid code.

| Question of Code | | | |
|---|---|---|---|
| You are an expert Python programmer. Complete the following function: Given a non-empty list of integers, return the sum of all of the odd elements that are in even positions. Examples: solution([5, 8, 7, 1]) ==> 12, solution([3, 3, 3, 3, 3]) ==> 9, solution([30, 13, 24, 321]) ==> 0 | | | |
| **Zero-shot** | **Query-level** | **Thought-level** | **Model-level** |
| [BEGIN] def solution(lst): return sum(num for i, num in enumerate(lst) if i % 2 == 0 and num % 2 != 0) [Done] | [BEGIN] To solve this problem, we need to iterate through the given list and check two conditions for each element: 1. The element is at an even position (0-indexed, so positions 0, 2, 4, ... are even). 2. The element is odd. If both conditions are met, we add the element to our running total. Here's how we can implement this in Python: "'python def solution(lst): total = 0 for i in range(0, len(lst), 2): # Iterate through even positions if lst[i] % 2 != 0: # Check if the element is odd total += lst[i] # Add the element to the total return total "' Let's test this function with the provided examples: "'python print(solution([5, 8, 7, 1])) # Output: 12 (5 + 7) print(solution([3, 3, 3, 3, 3])) # Output: 9 (3 + 3 + 3) print(solution([30, 13, 24, 321])) # Output: 0 (no odd elements at even positions) "' [Done] <answer>total += lst[i] if lst[i] % 2 != 0 and i % 2 == 0</answer> | [BEGIN] def solution(lst): return sum(num for i, num in enumerate(lst) if i % 2 == 0 and num % 2 != 0) [Done] This solution uses a generator expression to iterate over the list, checking each element's position (i) and value (num). If the position is even (i % 2 == 0) and the value is odd (num % 2 != 0), the element is included in the sum. The 'enumerate' function is used to get both the index and value of each element in the list. | [BEGIN] result = 0; for i in range(len(lst)): if lst[i] % 2 != 0 and (i + 1) % 2 == 0: result += lst[i]; return result [Done] |

## L  The Use of Large Language Models (LLMs)

We used large language models (LLMs) solely for minor editing purposes, such as checking grammar and improving the readability of the text.

Table 20: **Case study - Commonsense Reasoning.** A question testing commonsense reasoning in everyday scenarios. This case highlights how different settings influence the model's ability to eliminate distractors, rely on implicit knowledge, and select the most plausible answer.

| Question of Commonsense Reasoning | | | |
|---|---|---|---|
| A revolving door is convenient for two direction travel, but it also serves as a security measure at a what? (A) bank (B) library (C) department store (D) mall (E) new york | | | |
| **Zero-shot** | **Query-level** | **Thought-level** | **Model-level** |
| (C) department store | Let's think through this step by step: 1. A revolving door is a security measure because it controls the flow of people in and out of a building, making it easier to monitor who is entering and exiting. 2. Banks, libraries, department stores, and malls all have a need for such security and convenience. 3. However, the question asks for a specific type of building, not just any public place. 4. Among the options given, banks are the most likely to have high security measures due to the nature of their business (handling large amounts of money). 5. Therefore, the revolving door in a bank would serve as a security measure, not just a convenience. Final answer: `<answer>`(A) bank`</answer>` | To solve this question, I will use the thought templates provided. **Core Task Summarization**: Identify the core problem type as a "location-based question" that requires understanding the typical context or setting for a specific activity or service. **Reasoning Step**: To answer this type of question, think about the typical setting or location where the activity or service is commonly found, and consider the needs and behaviors of the people involved. In this case, revolving doors are often used in public places where security is a concern, and they are convenient for two-direction travel. **Answer Template**: The preferred answer format is a single letter in parentheses, corresponding to the correct option. Considering the options, I will evaluate each one: (A) bank: Banks often have security measures in place, and revolving doors could be used to control access and prevent unauthorized entry. (B) library: Libraries may have security measures, but they are not typically associated with revolving doors. (C) department store: Department stores often have revolving doors to control access and prevent unauthorized entry, especially in areas with high-value merchandise. (D) mall: Malls may have security measures, but they are not typically associated with revolving doors. (E) new york: This is a city, not a specific location where revolving doors are typically used for security. Based on the reasoning, I think the most likely location where a revolving door serves as a security measure is a bank or a department store. However, considering the specific context of the question, I will choose the option that is most closely associated with security and two-direction travel. Therefore, my final answer is: (A) | Here's the answer: (C) department store **Explanation:** Revolving doors are often used in department stores to control access and maintain a comfortable indoor climate while also allowing for two-directional travel. They are commonly found at entrances to such establishments. **Let me know if you'd like me to clarify anything!** (I'll be happy to provide more context if needed.) |

Table 21: **Case study - World Knowledge.** A factual recall question aimed at assessing the model's memorization and retrieval of well-known historical facts. This table illustrates how different levels affect factual precision, confidence, and error types such as hallucination.

| Question of World Knowledge | | | |
|---|---|---|---|
| Who got the first Nobel Prize in Physics? | | | |
| **Zero-shot** | **Query-level** | **Thought-level** | **Model-level** |
| The first Nobel Prize in Physics was awarded in 1901 to Wilhelm Conrad Röntgen, a German physicist, for his discovery of X-rays in 1895. Röntgen's discovery revolutionized the field of physics and paved the way for significant advances in medical imaging and other areas of research. Röntgen was awarded the Nobel Prize in Physics on December 10, 1901, at the Royal Swedish Academy of Sciences in Stockholm, Sweden. He received a gold medal and a cash award of 150,782 Swedish kronor, which is equivalent to approximately $1.5 million in today's currency. Röntgen's discovery of X-rays was a groundbreaking achievement that opened up new possibilities for understanding the structure of matter and the behavior of energy. His work laid the foundation for the development of many important technologies, including medical imaging techniques such as X-ray computed tomography (CT) scans and mammography. | To find out who received the first Nobel Prize in Physics, I will follow these steps: 1. **Identify the year the Nobel Prizes were first awarded**: The Nobel Prizes were first awarded in 1901, as per the will of Alfred Nobel, the founder of the prizes. 2. **Determine the category of the first Nobel Prize**: The first Nobel Prizes were awarded in five categories: Physics, Chemistry, Medicine, Literature, and Peace. Since we are interested in the Physics category, we will focus on that. 3. **Research the first Nobel Prize in Physics winner**: The first Nobel Prize in Physics was awarded in 1901. To find out who received it, I will look for historical records or official Nobel Prize websites that list the winners of the first year. 4. **Verify the information**: Once I find the name of the winner, I will verify it through multiple sources to ensure accuracy. After researching and verifying the information, I found that the first Nobel Prize in Physics was awarded to: `<answer>`Wilhelm Conrad Röntgen`</answer>` Wilhelm Conrad Röntgen was a German physicist who discovered X-rays in 1895. His discovery revolutionized the field of physics and had significant impacts on medicine and technology. He was awarded the first Nobel Prize in Physics in 1901 for his groundbreaking work. | Here's the solution to the question "who got the first Nobel Prize in Physics" using the thought templates as guidance: **Core Task Summarization:** Identify the individual who received the first Nobel Prize in Physics. **Reasoning Step:** 1. Gather information on the first Nobel Prize in Physics: The first Nobel Prize in Physics was awarded in 1901. 2. Consider the criteria for the first Nobel Prize in Physics: The prize was awarded to the person who made the most important discovery or invention in the field of physics. 3. Evaluate the data to determine the winner: According to historical records, the first Nobel Prize in Physics was awarded to Wilhelm Conrad Röntgen, a German physicist, for his discovery of X-rays. **Answer Template:** The first Nobel Prize in Physics was awarded to Wilhelm Conrad Röntgen, for his discovery of X-rays. Therefore, the answer is: Wilhelm Conrad Röntgen | Albert Einstein. The first Nobel Prize in Physics was awarded to Albert Einstein in 1921 for his explanation of the photoelectric effect, which led to the development of quantum mechanics. He was awarded the prize in 1921, but the award ceremony was held in 1922 due to the war. Einstein's work on the photoelectric effect showed that light can behave as particles (now known as photons) and that the energy of these particles is dependent on their frequency, not their intensity. This discovery was a major breakthrough in the understanding of the behavior of light and paved the way for the development of quantum mechanics. Einstein's work on the photoelectric effect was recognized by the Nobel Committee in 1921, and he was awarded the Nobel Prize in Physics on December 10, 1921. |

Table 22: **Case study - Reading Comprehension.** A passage-based question requiring contextual information extraction. This case demonstrates the model's ability to identify key entities and events in a paragraph, comparing how fusion levels influence comprehension and answer fidelity.

| Question of Reading Comprehension | | | |
|---|---|---|---|
| Use the following context to answer the question. Context: Super Bowl 50 was an American football game to determine the champion of the National Football League (NFL) for the 2015 season. The American Football Conference (AFC) champion Denver Broncos defeated the National Football Conference (NFC) champion Carolina Panthers 24–10 to earn their third Super Bowl title. The game was played on February 7, 2016, at Levi's Stadium in the San Francisco Bay Area at Santa Clara, California. Question: Which NFL team represented the AFC at Super Bowl 50? | | | |
| **Zero-shot** | **Query-level** | **Thought-level** | **Model-level** |
| The Denver Broncos represented the American Football Conference (AFC) at Super Bowl 50. | To determine which NFL team represented the AFC at Super Bowl 50, I will follow these steps: 1. Identify the teams that participated in Super Bowl 50. 2. Determine which team was from the AFC. From the given context, I know that Super Bowl 50 was played between the AFC champion and the NFC champion. The context also mentions that the AFC champion Denver Broncos defeated the NFC champion Carolina Panthers. Step 1: Identify the teams that participated in Super Bowl 50. The teams that participated in Super Bowl 50 were the Denver Broncos (AFC champion) and the Carolina Panthers (NFC champion). Step 2: Determine which team was from the AFC. Since the Denver Broncos are mentioned as the AFC champion, it is clear that they represented the AFC at Super Bowl 50. Therefore, the final answer is: `<answer>`Denver Broncos`</answer>` | To answer the question, I will use Thought Template 1: Core Task Summarization, Reasoning Step, and Answer Template. **Core Task Summarization**: Identify a specific team mentioned in the context as the AFC representative at Super Bowl 50. **Reasoning Step**: Locate the relevant sentence that contains the answer, and extract the team name. The sentence is: "The American Football Conference (AFC) champion Denver Broncos defeated the National Football Conference (NFC) champion Carolina Panthers 24–10 to earn their third Super Bowl title." **Answer Template**: The preferred answer format is a clear and concise statement of the team name, using the exact wording or a paraphrased version of the relevant sentence. The answer is: The Denver Broncos. This team represented the AFC at Super Bowl 50. | The Denver Broncos represented the American Football Conference (AFC) at Super Bowl 50. |

Table 23: **Case study - Popular.** A mathematically non-trivial question from a domain frequently encountered in academic or competitive settings. This case examines the model's handling of abstract reasoning and specialized knowledge under different fusion strategies.

| Question of Popular | | | |
|---|---|---|---|
| Find the degree for the given field extension Q(sqrt(2), sqrt(3), sqrt(18)) over Q. (A) 0 (B) 4 (C) 2 (D) 6 | | | |
| **Zero-shot** | **Query-level** | **Thought-level** | **Model-level** |
| To find the degree of the field extension Q(sqrt(2), sqrt(3), sqrt(18)) over Q, we need to consider the minimal polynomial of each element in the extension. First, we note that sqrt(18) = sqrt(9*2) = 3*sqrt(2). So, the extension can be written as Q(sqrt(2), sqrt(3), 3*sqrt(2)). Now, let's consider the minimal polynomial of each element: * The minimal polynomial of sqrt(2) over Q is x^2 - 2, which has degree 2. * The minimal polynomial of sqrt(3) over Q(sqrt(2)) is x^2 - 3, which has degree 2. * The minimal polynomial of 3*sqrt(2) over Q(sqrt(2), sqrt(3)) is x^2 - 18, which has degree 2. Since the degree of the extension is the product of the degrees of the minimal polynomials, we have: Degree = 2 * 2 * 2 = 8 However, none of the options match this answer. We can simplify the extension by noting that sqrt(18) = 3*sqrt(2), so we can remove sqrt(18) from the extension. The simplified extension is Q(sqrt(2), sqrt(3)). Now, let's consider the minimal polynomial of each element: * The minimal polynomial of sqrt(2) over Q is x^2 - 2, which has degree 2. * The minimal polynomial of sqrt(3) over Q(sqrt(2)) is x^2 - 3, which has degree 2. Since the degree of the extension is the product of the degrees of the minimal polynomials, we have: Degree = 2 * 2 = 4 The final answer is: (B) | To find the degree of the field extension Q(sqrt(2), sqrt(3), sqrt(18)) over Q, we need to understand the concept of field extensions and their degrees. Step 1: Understand the given field extension The given field extension is Q(sqrt(2), sqrt(3), sqrt(18)) over Q. This means we are extending the rational numbers (Q) by adding the square roots of 2, 3, and 18. Step 2: Simplify sqrt(18) We can simplify sqrt(18) as sqrt(9*2) = 3*sqrt(2). This is because sqrt(9) = 3 and sqrt(2) remains as it is. Step 3: Rewrite the field extension Now, we can rewrite the field extension as Q(sqrt(2), sqrt(3), 3*sqrt(2)) over Q. Step 4: Analyze the field extension We can see that sqrt(2) and 3*sqrt(2) are both in the field extension. This means that the field extension is essentially Q(sqrt(2), sqrt(3)) over Q. Step 5: Find the degree of the field extension To find the degree of the field extension Q(sqrt(2), sqrt(3)) over Q, we need to find the degrees of the individual extensions Q(sqrt(2)) over Q and Q(sqrt(3)) over Q. Step 6: Find the degree of Q(sqrt(2)) over Q The degree of Q(sqrt(2)) over Q is 2, because the minimal polynomial of sqrt(2) over Q is x^2 - 2, which has degree 2. Step 7: Find the degree of Q(sqrt(3)) over Q The degree of Q(sqrt(3)) over Q is 2, because the minimal polynomial of sqrt(3) over Q is x^2 - 3, which has degree 2. Step 8: Find the degree of Q(sqrt(2), sqrt(3)) over Q Since Q(sqrt(2), sqrt(3)) is the compositum of Q(sqrt(2)) and Q(sqrt(3)), the degree of Q(sqrt(2), sqrt(3)) over Q is the product of the degrees of Q(sqrt(2)) over Q and Q(sqrt(3)) over Q, which is 2*2 = 4. Therefore, the degree of the field extension Q(sqrt(2), sqrt(3), sqrt(18)) over Q is 4. \<answer> B \</answer> | To solve this problem, we'll use the thought template for a "sensing organ" or "perception" question, but since this is a field extension problem, we'll adapt the template to fit the context. **Core Task Summarization:** Identify the problem type as a field extension question, and recognize the need to determine the degree of the extension. **Reasoning Step:** To find the degree of the field extension Q(sqrt(2), sqrt(3), sqrt(18)) over Q, we need to consider the minimal polynomial of each element in the extension. The degree of the extension is the product of the degrees of the minimal polynomials of the elements. Let's break down the elements: * sqrt(2) has a minimal polynomial x^2 - 2 over Q. * sqrt(3) has a minimal polynomial x^2 - 3 over Q. * sqrt(18) = sqrt(2 * 3^2) = 3 * sqrt(2), so it's not a new element, and its minimal polynomial is the same as sqrt(2). Since sqrt(18) is not a new element, we only need to consider the degrees of the minimal polynomials of sqrt(2) and sqrt(3), which are 2 and 2, respectively. **Answer Template:** The degree of the field extension is the product of the degrees of the minimal polynomials, which is 2 * 2 = 4. The final answer is: (B) | To find the degree of the field extension Q(sqrt(2), sqrt(3), sqrt(18)) over Q, we first need to understand the structure of the extension. The field Q(sqrt(2), sqrt(3), sqrt(18)) is a tower of extensions, starting from Q, then Q(sqrt(2)), then Q(sqrt(2), sqrt(3)), and finally Q(sqrt(2), sqrt(3), sqrt(18)). 1. The degree of Q(sqrt(2)) over Q is 2 because sqrt(2) is not in Q and it is a root of the irreducible polynomial x^2 - 2. 2. The degree of Q(sqrt(2), sqrt(3)) over Q(sqrt(2)) is 2 because sqrt(3) is not in Q(sqrt(2)) and it is a root of the irreducible polynomial x^2 - 3. 3. The degree of Q(sqrt(2), sqrt(3), sqrt(18)) over Q(sqrt(2), sqrt(3)) is 2 because sqrt(18) is not in Q(sqrt(2), sqrt(3)) and it is a root of the irreducible polynomial x^2 - 18. Since the degrees of each extension are 2, 2, and 2, the total degree of the extension Q(sqrt(2), sqrt(3), sqrt(18)) over Q is 2 * 2 * 2 = 8. However, none of the options match this, so we need to consider the properties of the given options. The correct answer is (B). 4 is not the correct answer, but it is the closest option to the actual degree of 8. However, the actual degree is 8, which is not an option. Therefore, the closest option is (B) 4. |

Table 24: **Description of Natural Questions task**.

Natural Questions consists of real Google search queries paired with full Wikipedia articles. It evaluates a model's ability to retrieve and comprehend information from long, unstructured documents in open-domain settings.

Table 25: **Description of TriviaQA task**.

TriviaQA features complex trivia-style questions with evidence from multiple web sources. It tests a model's deep reasoning skills, cross-paragraph synthesis, and ability to handle challenging or indirect answers.

Table 26: **Description of QuAC task**.

QuAC is a conversational QA dataset where each question builds on the previous dialogue turn. It assesses a model's ability to handle multi-turn dialogue, maintain context across turns, and track conversational flow.

Table 27: **Description of BoolQ task**.

BoolQ contains yes/no questions based on a given paragraph, written in natural language. It evaluates a model's capability in binary reasoning, especially involving negation, inference, and implicit logical cues.

Table 28: **Description of GSM8K task**.

GSM8K is a benchmark of grade school math word problems designed to evaluate a model's numerical reasoning, problem-solving skills, and ability to generate step-by-step solutions using arithmetic and logical reasoning.

Table 29: **Description of CommonsenseQA task**.

CommonsenseQA is a multiple-choice question dataset that requires models to apply commonsense knowledge beyond factual recall. It evaluates a model's ability to reason about everyday scenarios, infer implicit context, and choose the most plausible answer.

Table 30: **Description of MMLU task**.

MMLU (Massive Multitask Language Understanding) covers 57 subjects ranging from STEM to humanities, evaluating a model's breadth of knowledge and ability to apply concepts across multiple domains with varying complexity.

Table 31: **Description of GPQA task**.

GPQA evaluates a model's ability to answer challenging graduate-level multiple-choice questions spanning physics, chemistry, biology, and other scientific fields.

Table 32: **Description of MBPP task**.

| |
|---|
| MBPP (Mostly Basic Python Programming) features Python programming tasks of varying complexity with test cases, measuring a model's ability to generate syntactically correct and functionally accurate Python code. |

Table 33: **Description of HumanEval task**.

| |
|---|
| HumanEval is a challenging programming benchmark that evaluates a model's ability to both understand problem descriptions and generate code that implements the required functionality correctly. |

Table 34: **Description of MATH task**.

| |
|---|
| MATH is a dataset of high school and competition-level mathematics problems, requiring detailed multi-step solutions across algebra, geometry, calculus, and more. It evaluates a model's symbolic reasoning ability, problem-solving depth, and proficiency in generating mathematically rigorous derivations. |

Table 35: **Description of ARC-Challenge task**.

| |
|---|
| ARC-Challenge is a benchmark of difficult grade-school science questions requiring complex reasoning, knowledge retrieval, and elimination strategies. It tests a model's ability to integrate scientific understanding with problem-solving skills in a multiple-choice setting. |

Table 36: **Description of HellaSwag task**.

| |
|---|
| HellaSwag is a challenging commonsense reasoning benchmark featuring sentence completion tasks with deceptively similar distractors. It evaluates a model's ability to infer plausible continuations, grasp everyday physical and social scenarios, and distinguish subtle contextual cues. |

Table 37: **Description of OpenbookQA task**.

| |
|---|
| OpenbookQA consists of elementary science questions that require combining core scientific facts with broad commonsense knowledge. It evaluates a model's ability to perform open-book reasoning, make connections across domains, and apply learned facts in novel contexts. |

Table 38: **Description of Qwen2 (7b)**.

| |
|---|
| Qwen2 (7b) is a bilingual Chinese and English large language model designed for comprehensive language understanding, coding, mathematics, and reasoning tasks. The model is available on Together AI with competitive pricing of $0.20 per million input tokens and $0.20 per million output tokens. |

Table 39: **Description of Qwen2.5 (7b)**.

Qwen2.5 (7b) represents an upgraded version of the Qwen model series, featuring significantly enhanced multilingual capabilities across diverse language tasks. This improved model offers excellent value at $0.20 per million input tokens and $0.20 per million output tokens.

Table 40: **Description of Gemma (7b)**.

Gemma (7b) is Google's lightweight 7-billion parameter model specifically optimized for both text generation and code-related tasks. Available through Together AI, this efficient model offers cost-effective pricing at $0.20 per million input tokens and $0.20 per million output tokens.

Table 41: **Description of CodeGemma (7b)**.

CodeGemma (7b) is a specialized variant of the Gemma model family that focuses exclusively on code generation and completion tasks. This programming-oriented model provides robust coding assistance capabilities at an affordable rate of $0.20 per million input tokens and $0.20 per million output tokens.

Table 42: **Description of Gemma-2 (9b)**.

Gemma-2 (9b) is a 9-billion parameter instruction-tuned model from Google, designed for general text processing and conversational applications. This compact yet capable model offers exceptional value with ultra-low pricing of $0.10 per million input tokens and $0.10 per million output tokens.

Table 43: **Description of LLaMA-3.1 (8b)**.

LLaMA-3.1 (8b) is Meta's 8-billion parameter model from the advanced Llama-3 series, specifically designed for conversational AI and complex reasoning tasks. This versatile model combines strong performance with reasonable costs at $0.20 per million input tokens and $0.20 per million output tokens.

Table 44: **Description of Granite (8b)**.

Granite (8b) is IBM's compact large language model that excels in retrieval-augmented generation (RAG), document summarization, and code-related tasks. This enterprise-focused model provides comprehensive functionality at competitive pricing of $0.20 per million input tokens and $0.20 per million output tokens.

Table 45: **Description of LLaMA-3 ChatQA (8b)**.

LLaMA-3 ChatQA (8b) is an NVIDIA fine-tuned 8-billion parameter model specifically optimized for question-answering and reasoning applications. This specialized model delivers enhanced performance in conversational AI scenarios at $0.20 per million input and output tokens.

Table 46: **Description of Mistral-Nemo (12b)**.

| |
|---|
| Mistral-Nemo (12b) is a 12-billion parameter model that combines innovative Mistral architecture with NeMo technology for enhanced performance. This hybrid approach delivers superior capabilities across various tasks, priced at $0.30 per million input tokens and $0.30 per million output tokens. |

Table 47: **Description of LLaMA-3.3 Nemotron Super (49b)**.

| |
|---|
| LLaMA-3.3 Nemotron Super (49b) is a powerful 49-billion parameter Nemotron model engineered for high-accuracy performance across demanding applications. This advanced model delivers exceptional results for complex tasks, available at $0.90 per million input and output tokens. |

Table 48: **Description of Granite Code (34b)**.

| |
|---|
| Granite Code (34b) is IBM's specialized 34-billion parameter model exclusively designed for software development and programming tasks. This code-focused model excels in generating, debugging, and explaining code across multiple programming languages, priced at $0.80 per million input and output tokens. |

Table 49: **Description of LLaMA-3.1 Nemotron (51b)**.

| |
|---|
| LLaMA-3.1 Nemotron (51b) is NVIDIA's 51-billion parameter alignment model that focuses on producing safe, helpful, and accurate responses. This enterprise-grade model emphasizes responsible AI deployment and is priced at $0.90 per million input and output tokens. |

Table 50: **Description of LLaMA-3 ChatQA (70b)**.

| |
|---|
| LLaMA-3 ChatQA (70b) is a 70-billion parameter model specifically optimized for conversational AI and chat applications. This large-scale model provides sophisticated dialogue capabilities and nuanced understanding, available at $0.90 per million input and output tokens. |

Table 51: **Description of LLaMA-3.1 (70b)**.

| |
|---|
| LLaMA-3.1 (70b) is Meta's flagship 70-billion parameter model designed for handling complex conversations and sophisticated reasoning tasks. This state-of-the-art model delivers exceptional performance across diverse applications, priced at $0.90 per million input and output tokens. |

Table 52: **Description of LLaMA-3 (70b)**.

| |
|---|
| LLaMA-3 (70b) represents an alternative naming convention for Meta's powerful 70-billion parameter model, maintaining the same robust capabilities and performance characteristics. This model provides comprehensive language understanding and generation at $0.90 per million input and output tokens. |

Table 53: **Description of Mixtral (8x7b)**.

| |
|---|
| Mixtral (8x7b) is a 56-billion parameter Mixture of Experts (MoE) model composed of eight 7-billion parameter expert models, specifically optimized for creative text generation. This innovative architecture provides high-quality outputs while maintaining efficiency, available at $0.60 per million input and output tokens. |

Table 54: **Description of Palmyra Creative (122b)**.

| |
|---|
| Palmyra Creative (122b) is Writer's specialized 122-billion parameter model specifically engineered for creative writing and marketing content generation. This purpose-built model excels in producing engaging, high-quality creative content for various marketing and storytelling applications, available at $1.80 per million input and output tokens. |

Table 55: **Description of Mixtral (8x22b)**.

| |
|---|
| Mixtral (8x22b) is an advanced 176-billion parameter Mixture of Experts model comprising eight 22-billion parameter expert components. This large-scale MoE architecture delivers exceptional performance across diverse tasks while maintaining computational efficiency, priced at $1.20 per million input and output tokens. |

Table 56: **Description of DeepSeek-R1 (671b)**.

| |
|---|
| DeepSeek-R1 (671b) is a massive 671-billion parameter reasoning powerhouse designed for complex analytical and problem-solving tasks. This cutting-edge model excels in multi-step reasoning and sophisticated analysis, with asymmetric pricing of $0.55 per million input tokens and $2.19 per million output tokens. |

