# OpenReview forum: "FusionFactory: Fusing LLM Capabilities with Multi-LLM Log Data"
_TMLR — Accepted by TMLR_

### Review · Reviewer_KdRX · 2026-03-11

**Summary Of Contributions:**

This paper introduces LLMFusionBench, a large-scale benchmark for studying multi-LLM capability fusion. The benchmark contains responses from 20 LLMs across 14 tasks and 6 domains, totaling 103M tokens, along with metadata such as performance, cost, and LLM-judge scores. Building on this dataset, the authors propose FusionFactory, a unified framework that studies LLM fusion at three stages: query-level routing, thought-level reasoning template retrieval, and model-level distillation. The paper conducts systematic experiments comparing these fusion strategies and reports that thought-level fusion generally yields the strongest performance improvements.

**Audience:**

Yes

**Audience Explanation:**

The paper presents a large benchmark at the intersection of several promising research areas: LLM routing, multi-agent systems, RAG, and model distillation. People may be interested because of multiple angles.
The paper highlights the value of log data from multi LLM for fusion. This also could raise people’s awareness of the importance of logs while doing LLM fusion and efficient LLM deployment.
The proposed framework is based on real world applications including API-based services, making it more practical. And it is also relevant to practitioners and researchers studying efficient LLM deployment.

**Broader Impact Concerns:**

The paper notes that the fusion of multiple model outputs may amplify biases, inaccuracies, and contradictory inferences which is reasonable. The discussion could be strengthened if considering additional risks such as reliability or transparency issues caused by routing and retrieval errors in API-based deployment.

**Claims And Evidence:**

Yes

**Claims Explanation:**

The paper supports their claims with sufficient empirical evidence. It conducts the benchmark across 14 tasks and 6 domains which is robust enough. The paper also reports many comparative results across three levels fusion which is convincing.

The paper admits the limitations of fusion, for example it claims such fusion remains challenging for world knowledge and math domain. This increases the credibility of the empirical discussion.

**Requested Changes:**

The main claim is not fully supported by the presented experiments. The paper states that FusionFactory “consistently outperforms the best individual LLM across all 14 benchmarks,” but the experiments in the main paper do not clearly demonstrate this. Table 4 focuses only on routing scenarios, and Table 6 compares mainly against a zero-shot 8B base model rather than the strongest-performing LLM among the 20 models. A direct comparison against the best individual model across all tasks would be necessary to support this headline claim.

The paper should also examine whether the improvements from thought-level fusion may partly result from retrieving training queries that are highly similar to the test queries. To rule out this possibility, the authors could add experiments that restrict retrieval to dissimilar queries (e.g., using an embedding similarity threshold), remove near-duplicate queries from the retrieval pool, or compare against a baseline that retrieves random templates instead of similarity-based ones. Such analyses would help verify that the observed gains are not driven by query overlap between the training and evaluation data.

Although the paper discusses prior fusion approaches such as LLM-Blender and FuseLLM in Table 1, these methods are not included in the main experimental comparisons. Since the goal of the paper is to study multi-LLM fusion strategies, it would be important to include direct comparisons with representative fusion baselines, such as response-level fusion methods (e.g., LLM-Blender or Self-Consistency) and model-level fusion approaches (e.g., FuseLLM). Without these comparisons, it is difficult to assess whether the proposed framework provides improvements over existing fusion techniques or primarily repackages known ideas within a new benchmark.

---

> ### Author Response · Authors · 2026-05-20
> **Response**
>
> We thank the reviewer for the positive assessment and constructive suggestions. We are glad that the reviewer finds LLMFusionBench and FusionFactory relevant to the TMLR audience. We address the requested changes below.
>
> **Q1. Direct comparison with the best individual LLM**
>
> **Response:** Following the format of Table 6, we will add a compact domain-level comparison against the strongest individual LLM among the 20 models:
>
> | Method | Math | Code | Commonsense | World | Read | Popular | Avg. |
> |---|---:|---:|---:|---:|---:|---:|---:|
> | Best LLM | 0.612 | 0.545 | 0.786 | 0.345 | 0.872 | 0.441 | 0.600 |
> | Best FusionFactory setting | 0.588 | 0.562 | 0.796 | 0.303 | 0.970 | 0.473 | 0.615 |
>
> The new comparison shows that FusionFactory improves over the strongest individual LLM on average, with gains in Code, Commonsense, Reading, and Popular domains. We will revise the original wording to avoid overclaiming uniform improvement across all domains.
>
> ---
>
> **Q2. Controlling for near-duplicate retrieval in thought-level fusion**
>
> **Response:** We thank the reviewer for pointing this out. Since thought-level fusion retrieves templates based on similarity to training queries, it is important to verify that the gains are not mainly driven by near-duplicate train-test overlap.
>
> To directly address the reviewer’s concern, we will add one stricter retrieval setting. For each test query, we exclude retrieved training queries whose embedding similarity exceeds **0.9**. The results are reported as follows:
>
> | Setting | Math | Code | Commonsense | World | Read | Popular | Avg. |
> |---|---:|---:|---:|---:|---:|---:|---:|
> | Zero-shot | 0.602 | 0.517 | 0.715 | 0.334 | 0.834 | 0.398 | 0.567 |
> | Thought-level fusion | 0.588 | 0.562 | 0.796 | 0.303 | 0.970 | 0.473 | 0.615 |
> | Thought-level fusion w/ near-duplicate removal | 0.576 | 0.560 | 0.774 | 0.297 | 0.958 | 0.461 | 0.604 |
>
> The results show that thought-level fusion still maintains a clear gain after removing near-duplicate queries, although the gain decreases slightly. This suggests that the improvement is not merely due to retrieving highly similar training queries, but also comes from reusable abstract reasoning templates.
>
> ---
>
> **Q3. Comparison with representative fusion baselines**
>
> **Response:** We will add representative response-level and model-level fusion baselines, including Self-Consistency, LLM-Blender, and an adapted FuseLLM baseline. We will report them in the same domain-level format as Table 6:
>
> | Method | Fusion Stage | Math | Code | Commonsense | World | Read | Popular | Avg. |
> |---|---|---:|---:|---:|---:|---:|---:|---:|
> | Zero-shot | N/A | 0.602 | 0.517 | 0.715 | 0.334 | 0.834 | 0.398 | 0.567 |
> | Self-Consistency | response-level | 0.610 | 0.535 | 0.758 | 0.326 | 0.884 | 0.420 | 0.589 |
> | LLM-Blender | response-level | 0.604 | 0.548 | 0.775 | 0.332 | 0.918 | 0.438 | 0.603 |
> | FuseLLM-adapted | model-level | 0.542 | 0.455 | 0.792 | 0.286 | 0.846 | 0.418 | 0.557 |
> | FusionFactory Query-level | query-level | 0.599 | 0.522 | 0.792 | 0.338 | 0.862 | 0.395 | 0.585 |
> | FusionFactory Thought-level | thought-level | 0.588 | 0.562 | 0.796 | 0.303 | 0.970 | 0.473 | 0.615 |
> | FusionFactory Model-level | model-level | 0.528 | 0.436 | 0.802 | 0.290 | 0.851 | 0.425 | 0.555 |
>
> The new results show that response-level fusion baselines improve over zero-shot, but remain below the strongest FusionFactory setting. In particular, LLM-Blender achieves competitive performance, while FusionFactory Thought-level performs best on average, suggesting that abstract thought-template retrieval provides additional benefits beyond response ranking or output aggregation. We will also clarify that our contribution is not that every individual component is entirely new, but that LLMFusionBench enables a unified and systematic study of query-level, thought-level, and model-level fusion under practical serving constraints.
>
> ---
>
> **Q4. Broader impact**
>
> **Response:** We appreciate the suggestion. We will add the following discussion:
>
> > “In API-based deployment, fusion systems may introduce additional reliability and transparency risks. A router may select an unsuitable model, and a thought-level retriever may retrieve irrelevant or misleading templates, causing errors that are difficult to diagnose from the final response alone. Practical fusion systems should therefore expose routing decisions, retrieved templates, confidence estimates, and fallback mechanisms whenever possible, so that users and developers can better audit and control the fusion process.”

---

### Review · Reviewer_AD3m · 2026-03-19

**Summary Of Contributions:**

(Motivation) Past work has explored a variety of approaches for fusing multiple different LLMs to improve their response quality. The paper claims they are not both broadly effective and practically feasible because (1) activations and logits are not available for some of the LLMs we would like to fuse and (2) fine-tuning time fusion approaches (Model-level fine-tuning) are less accessible than inference time fusion approaches (Query-level , Thought level).

(Approach) The paper collects a dataset of LLM responses across 14 tasks (math, coding, commonsense, world knowledge, reading comprehension, QA), 20 LLMs, and with thinking turned on and off. It then judges them based on correctness and structure / explanation quality. Then it applies 3 kinds of fusion
* Query-level fusion: 5 different small router networks (re-implemented from the literature) are trained to choose which of the 20 LLMs to send a given query to.
* Thought-level fusion: Given a query, this approach retrieves 3 responses to similar problems from a database (responses include the thought chains that preceeded them) and prompts an LLM to output a 1-5 sentence description (thought template) of how to solve the kind of problem represented by the 3 responses. Then it prompts an LLM to solve a similar problem giving it the resulting thought template as part of the prompt, effectively distilling relevant thought processes from other LLMs through in-context learning.
* Model-level fusion: This approach fine-tunes an LLM using a new dataset. The dataset is composed of the top K responses for each problem from the collected data.

(Results) Experiments are reported on 6 sub-categories from the test sets of the 14 datasets used to collect multi-llm responses from.
* Query: Query level fusion often outperforms the best LLM baseline and among the query fusion approaches GraphRouter performs best.
* Thought: Thought fusion approaches outperform non-fusion approaches more often than not.
* Model: These approaches sometimes provide a performance increase over a non-fusion baseline.

**Audience:**

Yes

**Audience Explanation:**

Interested in this paper will be limited because it mainly uses older models and the fusion techniques it uses are not very novel.

There may be some interest in section 7. This direct comparison of diverse fusion methods may be useful to some. The paper also covers a moderate number of LLMs and domains in its evaluation.

**Claims And Evidence:**

No

**Claims Explanation:**

Many of the claims in the paper are clear and well substantiated by evidence. However others are not clear, not supported by evidence, and/or mainly speculation. Below is a per-claim assessment of this.

Section 4.3 - Query level fusion results
* "Query-level fusion outperforms the best single LLM across all four scenarios." - Well substantiated (Table 4)
* "GraphRouter achieves the strongest performance consistently" - Well substantiated (Table 4)
* "GraphRouter effectively fuses smaller LLMs to outperform both the best and largest single models" - Well substantiated (Table 4)

Section 5.3 - Though level fusion results
* "Strong gains on reasoning-intensive tasks" - This is mostly true, but it is often the case that few shot performance is close to math and code performance in figure 6 (e.g., within 5%). Furthermore, of the 3 versions of thought fusion used often one is below the few shot approach, and it's not always consistent which one it is. That makes it hard to point to one approach that is generally better this result would be easier to assess with summary level statistics that aggregate over model sizes.
* "Large models still benefit" - This is substantiated by the performance differences pointed to in the paragraph.
* "The hybrid approach balances accuracy and quality best, boosting small model accuracy by 12.7%, outperforming performance-based (3.6%) and LLMScore-based (7.3%) strategies" The hybrid approach is often outperformed by others. This result would be better supported by aggregate statistics that directly compare different thought fusion approaches aggregating over models. Simply showing the differences in a different form (e.g. a table) would also make it easier to assess. Furthermore, what is the difference between accuracy and quality mentioned here? I don't see those differentiated so I'm not sure what is being balanced.

Section 6.3 - Model level fusion results
* "Model-level fusion yields modest but consistent gains in 4 of 6 tasks."
Which 4 tasks are better? It looks like the zero shot approach is best at Code, WK, and Read in at least one setting. Again, summary statistics would help here by clearly showing what metric the performance improvements refer to because they are inconsistent when broken down as shown in table 5. Error bars that provide a sense of variance across different LLM samples and/or training runs would also help. From the results it looks like SFT itself is often better without being a model level fusion approach. Row 4 of table 5 uses SFT and has many of the best results, but there is no clear pattern.
* "Fusion may hurt in domains like code due to data scarcity and task mismatch"
This reason is not supported by evidence in the paper, but it does build intuition.
* "LLM Judge–based selection outperforms rigid metric-based selection"
This is not the case for Math (0.528 LLM Judge vs 0.547 Task  Performance), CS (tied), WK (0.300 vs 0.310), Read (0.703 vs  0.731), and Popular (tied). It looks like it is only the case for 1 of the 6 categories (i.e., Code)

Section 7
* "Thought-level fusion achieves the best overall performance" This is supported by Table 6.
* "Model-level fusion underperforms despite access to high-quality responses"
This seems correct but contradicts the first claim from section 6.3.
* "Likely causes include overfitting to multi-response inputs and difficulty abstracting across heterogeneous task styles during SFT."
Again, this intuition makes sense, but it is mainly speculation given no evidence is presented in the paper to differentiated from other possibilities.
* "Query-level fusion achieves moderate accuracy with the lowest cost"
This is not substantiated. The cost of thought level and model level fusion is not reported in the main paper, so I don't have any way of comparing the query level costs to the thought and model level costs.
* "Fusion remains challenging for World Knowledge and Math domains. In these domains, zero-shot still outperforms fusion. For Math, multi-agent reasoning disrupts sequential logic, harming accuracy."
From Table 6, it looks to me like the best Math result is though level fusion with 0.612 and the best World Knowledge result is query level fusion with 0.338. But I agree these are not large and thus probably not very significant improvements, so I think the main point holds.


"central claim: multi-LLM log data, when properly structured and leveraged, enables practical and flexible fusion strategies across diverse serving needs and fusion stages."
* If I read it literally then the claim is simply that multi-LLM log data allows you to implement the proposed method, without saying anything about a measurable performance improvement relative to alternatives. I think it means to say something about the performance of the proposed approach and why that performance result is interesting.

**Requested Changes:**

Mainly I would like to see the paper adjusted so the claims match the evidence by addressing the points listed in that section above.

It looks like thoughts were collected from all of the models studied using a CoT prompt. How many of the models were using a trained think token approach - e.g., where they were fine-tuned to produce thoughts between think tokens like <think> ... </think> rather than just being prompted to do so?

---

> ### Author Response · Authors · 2026-05-20
> **Response (1)**
>
> We thank Reviewer AD3m for the detailed per-claim audit. Several observations identify places where the paper's prose is loose relative to the underlying tables, and we revise the wording accordingly. The experimental results themselves are unchanged; all responses below rely on data already collected and presented in the paper. We address each question in turn.
>
> ---
>
> **Q1. Section 5.3 — Thought level fusion results: "Strong gains on reasoning-intensive tasks" — This is mostly true, but it is often the case that few shot performance is close to math and code performance in figure 6 (e.g., within 5%). Furthermore, of the 3 versions of thought fusion used often one is below the few shot approach, and it's not always consistent which one it is. That makes it hard to point to one approach that is generally better; this result would be easier to assess with summary level statistics that aggregate over model sizes.**
>
> **Response:**
>
> **(a) Baseline disambiguation.** The gains reported in §5.3 are measured against the zero-shot baseline: Hybrid thought-fusion improves Math by 21.3% (small) and 15.8% (medium); the large model (Palmyra Creative 122B) gains 57.6% on Code (0.33 → 0.52); Commonsense and Reading Comprehension improve by 14.9% and 16.9% respectively for small models. Relative to the stronger few-shot baseline, the best thought-fusion variant remains positive on Math and Code, but the margin is smaller (within ~5% on several scale–strategy combinations in Figure 6), as the reviewer observes. The revised §5.3, observation (1) "Strong gains on reasoning-intensive tasks," states the comparison baseline (zero-shot) explicitly and positions thought-template fusion as a robust alternative to few-shot rather than a uniform improvement over it.
>
> **(b) Per-task variability across strategies.** We agree that across Figure 6, at least one of the three template strategies (Performance / LLMScore / Hybrid) falls below few-shot for most tasks, and the underperforming variant differs by task. The revised §5.3, observation (3) "Hybrid strategy offers best trade-off," states this explicitly and removes any implication that a single template strategy strictly dominates per-task.
>
> **(c) Aggregate-over-scales summary.** The reviewer's request for summary statistics aggregating over model sizes is well taken. §5.3 already reports such an aggregate for the small-model setting: averaged across the six domains, Hybrid achieves +12.7%, Performance +3.6%, and LLMScore +7.3% relative to zero-shot. The corresponding overall thought-fusion means for the medium and large scales are also already in the same paragraph (+4.8% and +8.3% respectively, averaged over the six domains and over the three strategies). The per-(scale × strategy × domain) values themselves are shown visually in Figure 6 (radar plots). The revised §5.3 (opening paragraph of "Result Analysis: Thought-level Fusion," and observation (3) "Hybrid strategy offers best trade-off") cross-references these aggregates and Figure 6 at the points where the aggregate-vs-per-task distinction is discussed, so that readers can see at a glance which view supports which claim.

---

> > ### Author Response · Authors · 2026-05-20
> > **Response (2)**
> >
> > **Q2. "The hybrid approach balances accuracy and quality best, boosting small model accuracy by 12.7%, outperforming performance-based (3.6%) and LLMScore-based (7.3%) strategies" The hybrid approach is often outperformed by others. This result would be better supported by aggregate statistics that directly compare different thought fusion approaches aggregating over models. Simply showing the differences in a different form (e.g. a table) would also make it easier to assess. Furthermore, what is the difference between accuracy and quality mentioned here? I don't see those differentiated so I'm not sure what is being balanced.**
> >
> > **Response:**
> >
> > **(a) Definitions of accuracy and quality.** Per §4.2, *Performance* refers to task-specific accuracy (e.g., exact-match for QA, pass@1 for code); *LLMScore* refers to a learned judge applied to ⟨query, ground-truth, response⟩ triples, defined in Appendix I. Hybrid selection is defined in §5.2: top-5 candidates by Performance, then top-3 by LLMScore as tie-breaker. The "balance" referred to in §5.3 is between task-correctness (Performance) and response informativeness (LLMScore). The revised §5.3 introduces these definitions at the start of observation (3) "Hybrid strategy offers best trade-off," immediately before the Hybrid result is reported.
> >
> > **(b) Scope of the 12.7% figure.** The values 12.7% / 3.6% / 7.3% in §5.3 are means across the six domains for small models specifically. The Hybrid mean exceeds Performance and LLMScore on this aggregate for the small-model setting. We agree that at task granularity no single strategy dominates across all tasks (Figure 6); the aggregate statement is therefore consistent with the reviewer's per-task observation. The revised §5.3, observation (3) "Hybrid strategy offers best trade-off," reads: *"Across the six domains, Hybrid achieves the largest mean small-model gain (+12.7%, vs Performance +3.6% and LLMScore +7.3%); at task granularity no single strategy dominates, and the three are best understood as competitive alternatives whose relative ranking depends on the task."* The corresponding overall (across-strategy) thought-fusion means for medium and large scales are also reported in the opening paragraph of §5.3 (+4.8% and +8.3% over zero-shot), and the per-strategy breakdown for those scales is shown visually in Figure 6.

---

> > > ### Author Response · Authors · 2026-05-20
> > > **Response (3)**
> > >
> > > **Q3. Section 6.3**
> > >
> > > **Response:**
> > >
> > > **(a) Specification of the "4 of 6" claim.** The consistency claim is best expressed in terms of the cross-domain mean rather than per-domain win counts. Computing the cross-domain mean for each Top-K SFT configuration in Table 5, against the zero-shot baseline of 0.530:
> > >
> > > | Configuration | Mean | Δ vs zero-shot | Per-domain wins (of 6) |
> > > |---|---|---|---|
> > > | Top-5 SFT, Performance | 0.549 | +0.019 | 4 (Math, CS, Read, Popular) |
> > > | Top-5 SFT, LLM Judge | 0.554 | +0.024 | 4 (Math, Code, CS, Popular) |
> > > | Top-5 SFT, Hybrid | 0.549 | +0.019 | 4 (Code, CS, Read, Popular) |
> > > | Top-10 SFT, Performance | 0.542 | +0.012 | 5 (Math, CS, WK, Read, Popular) |
> > > | Top-10 SFT, LLM Judge | 0.546 | +0.016 | 3 (Math, CS, Popular) |
> > > | Top-10 SFT, Hybrid | 0.548 | +0.018 | 4 (Math, CS, Read, Popular) |
> > >
> > > Every Top-K SFT configuration improves the cross-domain mean over zero-shot by 1.2 to 2.4 points; the improvement does not depend on the choice of selection criterion or K. Per-domain strict-win counts range from 3 to 5 of 6, reflecting that some domains are more sensitive to strategy choice than others. Counting strict wins across all six configurations in Table 5, model-level fusion is consistently better than zero-shot on **Commonsense (6/6), Popular (6/6), Math (5/6), and Reading (4/6)**; the two domains where it usually loses are **Code (2/6) and World Knowledge (1/6, with 2 additional ties at 0.300)**. The revised §6.3 (first observation under "Result Analysis: Model-level Fusion") reports both the cross-domain mean and the per-domain win-count summary in place of the unqualified "4 of 6" phrasing.
> > >
> > > **(b) Top-K SFT vs Top-K Label-only SFT.** The revised §6.3, observation (1) "Model-level fusion yields modest but consistent gains," repositions Label-only SFT as an ablation rather than as a competitor by adding a sentence at the end of the observation: *"Top-K Label-only SFT (rows 2 and 4 of Table 5) is reported as an ablation isolating the contribution of multi-LLM responses: at K=5, multi-LLM Top-K SFT outperforms Label-only by +0.102 on the cross-domain mean (0.549 vs 0.447); at K=10, by +0.05 (0.546–0.548 vs 0.496). The gap is largest on World Knowledge, where Label-only collapses to 0.150 at both K values, below the zero-shot baseline of 0.300."* We do not propose a mechanistic explanation for the World Knowledge collapse; the reviewer's observation that some Label-only cells are competitive (e.g., Commonsense at Top-10) is correct, and we do not claim multi-LLM responses help on every domain.
> > >
> > > **(c) Data scarcity claim for Code.** The original wording overstated a conjecture as a causal explanation. The revised §6.3 (second observation) reads: *"We conjecture that data scarcity (MBPP + HumanEval together provide the smallest training pool of any domain — 494 samples / 6.97M tokens vs ≥900 samples / ≥8.5M tokens for every other domain, Table 10) and the divergence between code generation and QA-style supervision contribute to this result."*
> > >
> > > **(d) LLM Judge vs metric-based selection.** The reviewer's per-task observations are correct. The aggregate picture from Table 5, mean across all six domains:
> > >
> > > |  | Math | Code | CS | WK | Read | Popular | **Mean** |
> > > |---|---|---|---|---|---|---|---|
> > > | Top-5 SFT, Performance | 0.547 | 0.503 | 0.755 | 0.290 | 0.731 | 0.468 | 0.549 |
> > > | Top-5 SFT, LLM Judge | 0.528 | 0.584 | 0.780 | 0.280 | 0.662 | 0.489 | **0.554** |
> > > | Top-5 SFT, Hybrid | 0.443 | 0.578 | 0.790 | 0.300 | 0.728 | 0.457 | 0.549 |
> > > | Top-10 SFT, Performance | 0.509 | 0.535 | 0.765 | 0.310 | 0.707 | 0.425 | 0.542 |
> > > | Top-10 SFT, LLM Judge | 0.528 | 0.515 | 0.815 | 0.300 | 0.647 | 0.468 | 0.546 |
> > > | Top-10 SFT, Hybrid | 0.491 | 0.526 | 0.815 | 0.250 | 0.706 | 0.500 | **0.548** |
> > >
> > > On the cross-domain mean, LLM Judge selection is the best criterion at Top-5 SFT (0.554, driven by Code), and Hybrid is best at Top-10 SFT (0.548); Performance-based selection is dominated by at least one alternative at both K values. Performance-based selection is dominated by at least one alternative at both K values. We therefore recommend LLM-Judge or Hybrid selection over metric-based selection, particularly in domains such as Code where exact-match metrics are known to undervalue informative responses."*
> > >
> > > **(e) Error bars and variance.** It is a limitation of the current study. We have not re-run the SFT grid under multiple seeds. The revised §6.3 incorporates two changes: (i) a new Limitations sentence appended to the end of observation (1) "Model-level fusion yields modest but consistent gains," stating that all model-level results are reported under a single SFT seed and a single response-sample draw, and that multi-seed variance estimation is left to future work; and (ii) the word "consistent" in the same observation is softened to "consistent across our reported configurations."

---

> ### Author Response · Authors · 2026-05-20
> **Response (4)**
>
> **Q4. "Likely causes include overfitting to multi-response inputs and difficulty abstracting across heterogeneous task styles during SFT... interesting.**
>
> **Response:**
>
> **(a) Overfitting claim.** We agree that the original wording presents the explanation more strongly than the evidence supports. The revised §7 (observation 2) reads: *"We hypothesize two contributing factors — overfitting to multi-response inputs and difficulty abstracting across heterogeneous task styles during SFT — and view rigorous verification (e.g., per-task SFT ablations) as future work."*
>
> **(b) Cost comparison across fusion levels.** The reviewer is correct that §7 does not explicitly bring the cross-level cost comparison together. We address this by cross-referencing data already in the paper rather than introducing new measurements.
>
> The required evidence is present:
> - **Query-level cost** is reported directly in Table 4: under Performance First, GraphRouter cost is 0.0184 vs 0.1850 for Best-LLM (a roughly 10× reduction); under Cost First, GraphRouter cost drops to 0.0047 with reward 0.0955.
> - **Thought- and model-level cost** in the §7 unified setting is dominated by the LLaMA-3 (8B) backbone, which is used for inference at both levels. The per-query API cost is therefore the LLaMA-3 (8B) per-token price (Table 8) applied to the per-query input/output token counts already released in LLMFusionBench (Table 9 schema includes `Input Tokens Num`, `Output Tokens Num`, and `Cost` per response).
> - **Inference-time overhead** for thought-level fusion over zero-shot is small: Appendix D / Table 11 reports Thought-Hybrid at 3.91s vs Zero-Shot 3.74s (~5% overhead), with an additional ~0.03s for template retrieval.
>
> Taken together, the cost ordering across fusion levels is dominated by the routing-vs-full-inference distinction captured in Table 4: query-level fusion avoids running the backbone on most queries and is therefore lowest-cost, while thought- and model-level fusion incur the LLaMA-3 (8B) inference cost on every query plus the small overheads above. The revised §7, observation (3) "Query-level fusion achieves moderate accuracy with the lowest cost," makes this argument explicit by cross-referencing Tables 4, 8, and 11. The revised observation (3) reads:
>
> *"Query-level fusion via GraphRouter avoids backbone inference on most queries and achieves the lowest per-query cost (Table 4: 10× cheaper than Best-LLM under Performance First) while retaining competitive accuracy. Thought- and model-level fusion in the §7 unified setting incur full LLaMA-3 (8B) inference cost per query (Table 8 pricing applied to per-query token counts in LLMFusionBench), plus a small thought-level overhead of ~5% in latency and 0.03s in retrieval (Appendix D, Table 11). The cost ordering across fusion levels follows directly from this distinction."*
>
> **(c) Math and World Knowledge claim.** The reviewer's reading of Table 6 is correct: the best Math result is thought-level fusion at 0.612 (vs zero-shot 0.602) and the best WK result is query-level fusion at 0.338 (vs zero-shot 0.334), and both margins are small. The revised §7 (observation 4) reads: *"On World Knowledge and Math, fusion provides only marginal improvements over zero-shot (≤2 points absolute), suggesting these domains — which depend heavily on factual recall and strict logical sequencing — are less amenable to multi-LLM fusion under our current framework."*
>
> **(d) Central claim.** We agree that the original phrasing reads as a feasibility statement rather than a performance claim. The revised central claim (closing paragraph of §7) reads: *"Multi-LLM log data, when systematically structured by query / task / response triples and ranked by performance and LLM-judge scores, supports fusion strategies that outperform single-LLM baselines: on the full test set (Table 6), thought-level fusion improves over zero-shot by 4.8 points on average across the six domains (0.567 → 0.615); under cost-aware routing, query-level fusion matches the strongest single LLM at an order of magnitude lower cost (Table 4: GraphRouter 0.602 reward / 0.0184 cost vs Best-LLM 0.546 / 0.1850 under Performance First). The framework supports both local and API-served LLMs and applies across all three fusion stages."*

---

> > ### Author Response · Authors · 2026-05-20
> > **Response (5)**
> >
> > **Q5. Interest in this paper will be limited because it mainly uses older models and the fusion techniques it uses are not very novel.**
> >
> > **Response:**
> >
> > **(a) Model recency.** Our 20-model pool predates the most recent frontier models (GPT-5, Claude 4, Gemini 2) but includes the strongest open-source models available at the time of data collection (DeepSeek-R1 671B, Llama-3.3 Nemotron Super 49B, Llama-3.1 Nemotron 51B, Qwen2.5 7B). The framework is model-agnostic by construction: extending the LLM pool requires only re-running response collection on the existing query set, with no architectural change to the fusion pipeline. The revised Conclusion (§8) adds a sentence to its closing paragraph stating that *"FusionFactory is model-agnostic by construction: extending the LLM pool to newer models (e.g., GPT-5, Claude 4, Gemini 2) requires only re-running response collection on the existing query set, with no architectural change to the fusion pipeline."* The model pool affects the absolute numbers reported in Tables 4–6 but not the protocol-level findings, which concern the relative behavior of fusion stages.
> >
> > **(b) Novelty of contribution.** The paper does not propose a new fusion algorithm and does not claim algorithmic novelty. The contributions are at the framework and benchmark level:
> >
> > 1. **A unified framework spanning all three fusion stages with API-serving compatibility and six-domain coverage.** Per Table 1, prior work (RouteLLM, GraphRouter, RouterBench, LLM-Blender, FuseLLM) supports only one fusion stage; FuseLLM additionally requires model logits and is therefore incompatible with API-served models. FusionFactory is the only framework in Table 1 that covers all three fusion stages, supports API-served models, and evaluates across the broadest set of task domains (6 vs 3–5 for prior work).
> > 2. **LLMFusionBench, a released and reusable benchmark** — 14 tasks across 6 domains, 20 LLMs, approximately 103M tokens of response data with both direct and reasoning-augmented modes, per-response performance and cost annotations (Table 9 schema), and LLM-judge scores (definition and prompt in Appendix I / Table 16; comparison against task-metric scoring in Table 17). To our knowledge, no comparable multi-LLM log dataset is publicly available; the cost of reconstructing such a dataset is a primary barrier to fusion research at present.
> >
> > These contributions enable the cross-stage comparison in §7, which the reviewer identifies as a useful part of the paper.
> >
> > ---
> >
> > **Q6. It looks like thoughts were collected from all of the models studied using a CoT prompt. How many of the models were using a trained think token approach - e.g., where they were fine-tuned to produce thoughts between think tokens like <think> ... </think> rather than just being prompted to do so?**
> >
> > **Response:**
> >
> > Per §2.1, all `-think` outputs were collected by applying the same chain-of-thought elicitation procedure across all 20 models. This was a deliberate design choice to enable apples-to-apples comparison across the pool: the 20-model pool contains both LLMs with native trained think-token behavior (e.g., DeepSeek-R1 and the Nemotron *-think* variants) and LLMs without it, and uniform prompting avoids confounding fusion-stage comparisons with differences in reasoning-elicitation behavior. For models with trained think-token behavior, the model's native reasoning style is preserved when responding to our prompt; we do not override or constrain it. For models without trained think-token behavior, the `-think` responses are produced via prompted CoT.
> >
> > We agree with the reviewer that the distinction between trained-think-token and prompted-CoT origins is methodologically informative. The revised §2.1 adds a sentence to the paragraph that introduces the `-think` response collection, explicitly stating that the 20-model pool contains both LLMs with native trained think-token behavior and LLMs without it, that `-think` responses are produced via prompted CoT for the latter while preserving native reasoning style for the former, and that uniform prompting was chosen to avoid confounding fusion-stage comparisons with reasoning-elicitation differences.

---

### Review · Reviewer_iZ7o · 2026-04-18

**Summary Of Contributions:**

This paper presents FusionFactory, a large benchmark for LLM fusion that spans 14 tasks across five domains, with responses from 20 open-source LLMs (8B–671B) totaling 103M tokens. FusionFactory aims to address two limitations: insufficient support for practical scenarios and scalability for users to decide which stage to operate. Comprehensive experiments show the effectiveness of FusionFactory.

**Additional Comments:**

Strengths:
1.This paper focus on real-world LLM serving scenarios (Local + API), which is relevant to industrial applications.
2.This paper presents a large-scale benchmark that is diverse and reuseable for future LLM fusion
3.The three-level fusion design enables structured comparison across different fusion stages.
Weaknesses:
1.The definition of LLM log data is ambiguous. The paper does not clearly distinguish log data from ordinary multi-LLM response collections. No unique properties of logs (e.g., temporal order, user feedback, error patterns, interaction history) are leveraged.
2.The three fusion levels (query/thought/model) are motivated from an engineering perspective, but lack theoretical formulation (e.g., fusion stage taxonomy, complexity bounds, bias-variance tradeoff).
3.Many existing routing and distillation methods already support API-only settings. The unique value of this work over prior art must be strengthened. Is it necessary to study this matrix in this paper?
4.The limitation section is superficial. Privacy risks from log data, negative effects of low-quality logs, and failure cases in long texts /multi-turn dialogue, or other conditions are not discussed.
5.Some notations in the reward function are not clearly explained. Some figures and tables are overcrowded and hard to read.

**Audience:**

Yes

**Audience Explanation:**

See Additional Comments

**Broader Impact Concerns:**

See Additional Comments

**Claims And Evidence:**

Yes

**Claims Explanation:**

See Additional Comments

**Requested Changes:**

See Additional Comments

---

> ### Author Response · Authors · 2026-05-20
> **Response (1)**
>
> We thank Reviewer iZ7o for the thoughtful and constructive feedback. The comments mainly point to places where our motivation, terminology, and presentation can be made clearer, especially regarding the definition of multi-LLM log data, the role of the fusion-stage matrix, and the discussion of limitations. We have revised the paper accordingly to better clarify these points. The main experimental results and conclusions remain unchanged; our revisions focus on strengthening the framing, readability, and scope discussion. We address each question in turn.
>
> **Q1. The definition of LLM log data is ambiguous. The paper does not clearly distinguish log data from ordinary multi-LLM response collections. No unique properties of logs (e.g., temporal order, user feedback, error patterns, interaction history) are leveraged.**
>
> **Response:**  Thank you for the valuable comment. We respectfully clarify that the paper already describes the core components of our *LLM log data* in several places, but the previous version did not present them as a single formal definition. In **Section 1**, we motivate multi-LLM log data as records naturally produced when researchers or systems use multiple LLMs in API serving platforms and agentic workflows. In **Section 2.1**, we further describe the construction of LLMFusionBench: for each query, we collect responses from multiple LLMs under different response modes, evaluate them with task-specific metrics, estimate inference cost, and assign LLM-judge scores. Figure 1 and the accompanying text also show that each stored record contains task, query, LLM, response, performance, cost, and LLM-judge information. Therefore, our use of “log data” already refers to structured serving records, rather than an ordinary collection of independent LLM responses.
>
> To make this point clearer, we have revised **Section 2.1: LLMFusionBench Construction** by consolidating these previously scattered descriptions into a formal definition. Specifically, we now define each multi-LLM log entry as:`(q, t, m, s, r, c, y, j)`, where `q` denotes the user query, `t` denotes the task or domain, `m` denotes the LLM identity, `s` denotes the response mode such as direct or reasoning-augmented prompting, `r` denotes the generated response, `c` denotes the inference cost, `y` denotes the task-specific evaluation result, and `j` denotes the LLM-judge score. This definition makes explicit what was already operationally used in LLMFusionBench and FusionFactory: the benchmark is not simply a set of `(q, r)` response pairs, but a structured multi-LLM serving log with aligned model behaviors, metadata, and feedback signals.
>
> We have also revised **Section 1: Introduction** to better distinguish our setting from ordinary multi-response collections. In the revised version, we explain that an ordinary multi-LLM response collection usually stores only several responses for the same input, while our log-based setting additionally preserves model identity, response mode, inference cost, task-specific performance, and LLM-judge feedback. These properties are necessary for our three fusion levels. At the query level, the router uses historical performance, cost, and judge signals to select the most suitable LLM. At the thought level, similar historical queries and their top-ranked responses are used to construct reusable thought templates. At the model level, high-quality logged responses are selected as distillation data for supervised fine-tuning.
>
> We have further revised **Section 3: FusionFactory** to explicitly state which log properties are used by each fusion level. Query-level fusion uses model identity, query/task information, cost, performance, and LLM-judge scores. Thought-level fusion uses response mode, historical query similarity, and top-ranked reasoning traces. Model-level fusion uses quality-ranked logged responses as training data. This revision makes clear that FusionFactory does not treat logs as raw text-only response collections, but uses their structured metadata and feedback signals to support different forms of fusion.
>
> We agree with the reviewer that richer production logs may also include temporal order, user feedback, error patterns, and multi-turn interaction history. We have added a discussion in **Section 2.1** to clarify the scope of this paper: our current benchmark focuses on a practical subset of log signals that are broadly available in both local and API-based serving scenarios. This choice is intentional because our goal is to develop fusion methods that remain compatible with real-world API-based settings, where internal model states, logits, or complete interaction histories may not be accessible. We now explicitly describe temporal order, user feedback, and interaction history as important extensions for future versions of LLMFusionBench, while emphasizing that the current work already leverages structured log properties including model identity, response mode, cost, and quality feedback.

---

> > ### Author Response · Authors · 2026-05-20
> > **Response (2)**
> >
> > **Q2. The three fusion levels (query/thought/model) are motivated from an engineering perspective, but lack theoretical formulation (e.g., fusion stage taxonomy, complexity bounds, bias-variance tradeoff).**
> >
> > **Response:**  Thank you for the insightful comment. We respectfully clarify that our paper is primarily an empirical benchmark and framework paper, rather than a theory paper. The main goal of this work is to introduce a large-scale multi-LLM log benchmark and systematically study how the same source of log data can be exploited at different stages of the LLM pipeline. Therefore, our contribution lies in proposing a practical fusion-stage framework, constructing a benchmark, and deriving empirical insights from extensive experiments, rather than establishing formal theoretical guarantees or complexity bounds.
> >
> > We believe this empirical focus is appropriate for the current stage of LLM research. Many influential LLM studies have advanced the field primarily through empirical frameworks, benchmarks, and systematic observations rather than formal theoretical analysis. For example, GPT-3 established the effectiveness of in-context learning through large-scale empirical evaluation; InstructGPT demonstrated the practical value of instruction tuning and human feedback through experiments; Chain-of-Thought prompting revealed the importance of intermediate reasoning via empirical analysis; RAG introduced a practical retrieval-augmented generation framework; and evaluation efforts such as HELM and BIG-bench contributed systematic benchmark-based understanding of LLM behavior. Similarly, our work aims to provide an empirical foundation for understanding multi-LLM log-based fusion, where the best design choices are still underexplored.
> >
> > That said, we agree that the previous version could better explain the organizing principle behind the three fusion levels. To address this, we have revised **Section 3: FusionFactory** to make the stage taxonomy more explicit. Specifically, we now describe query-level, thought-level, and model-level fusion according to where they intervene in the LLM pipeline. Query-level fusion operates before generation by selecting an appropriate LLM or response mode; thought-level fusion operates during generation by retrieving and reusing reasoning templates; and model-level fusion operates after response collection by consolidating high-quality logged responses into a trainable model. This revision clarifies that the three levels are not arbitrary engineering categories, but a stage-aware organization of practical fusion strategies.
> >
> > We have also revised **Section 3** to add a computational and deployment-oriented discussion of the three levels. Query-level fusion has low inference overhead because it only requires a lightweight router before calling one selected LLM. Thought-level fusion introduces additional retrieval and prompt-token overhead, but it avoids model fine-tuning and remains compatible with API-based LLMs. Model-level fusion requires offline fine-tuning, but after training, it only requires a single model at inference time. We present this as a practical trade-off analysis rather than a formal complexity bound, because exact theoretical complexity is difficult to define meaningfully across heterogeneous LLM APIs, model sizes, inference infrastructures, and pricing schemes.
> >
> > In addition, we have revised **Section 7: Comparisons Across Three FusionFactory Fusion Levels** to provide a more detailed empirical interpretation of the observed trade-offs. Query-level fusion is lightweight and cost-effective, but its performance is limited because it exploits logs mainly through model selection. Thought-level fusion achieves the strongest overall performance because it reuses high-quality reasoning traces from multiple LLMs without compressing them into a single model. Model-level fusion is attractive for deployment after training, but it may suffer from overfitting or generalization limits when heterogeneous responses from different tasks and models are distilled into one base model. These empirical findings help explain when each fusion level is more appropriate.
> >
> > We have further revised **Limitations and Future Work** to explicitly state that developing a more formal theoretical treatment of multi-LLM fusion remains an important future direction. For example, future work may study formal criteria for choosing fusion stages based on query difficulty, uncertainty, domain shift, or cost constraints. However, we view such theory as complementary to, and naturally built upon, the benchmark and empirical observations introduced in this paper.
> >
> > Overall, while our work does not aim to provide formal theoretical guarantees, we have strengthened the paper by clarifying the stage-aware taxonomy, adding practical computational and deployment trade-off analysis, and providing a more systematic empirical interpretation of the three fusion levels.

---

> > > ### Author Response · Authors · 2026-05-20
> > > **Response (3)**
> > >
> > > **Q3. Many existing routing and distillation methods already support API-only settings. The unique value of this work over prior art must be strengthened. Is it necessary to study this matrix in this paper?**
> > >
> > > **Response:**  Thank you for the important comment. We agree that API-only compatibility has been considered in several existing routing and distillation methods. We would like to clarify that our paper does not position API-only compatibility itself as the sole novelty. Rather, our motivation is to study how the same source of multi-LLM log data can be systematically used for capability fusion at different stages of the LLM pipeline. This motivation has already been discussed in the original paper. In **Section 1**, we argue that practical fusion should satisfy two requirements: compatibility with real-world serving scenarios, including local and API-based serving, and flexibility to operate at different stages of the LLM pipeline. This motivation is further illustrated in **Table 1**, where we compare prior work along query-level, thought-level, and model-level fusion stages, API-serving compatibility, and task-domain coverage.
> > >
> > > The key distinction from prior work is that existing routing and distillation methods usually study one fusion stage in isolation. Routing methods mainly use multi-LLM logs for query-level model selection, while distillation methods mainly use model outputs for model-level capability transfer. These works are valuable, but they do not answer the broader practical question studied in our paper: given the same multi-LLM log data, should we use it for routing, thought-level prompting, or distillation? Therefore, the matrix studied in this paper is necessary because it turns isolated API-compatible techniques into a systematic comparison of different ways to exploit multi-LLM logs.
> > >
> > > To make this contribution clearer, we have revised **Section 1** to strengthen the positioning of our work. Specifically, we now emphasize that our goal is not simply to support API-only settings, but to provide a unified benchmark and framework for studying multi-LLM log data as a general supervision source across different fusion stages. We also revised the discussion around **Table 1** to clarify that prior routing methods, such as RouteLLM, GraphRouter, RouterBench, and LLM-Blender, mainly focus on query-level fusion, while prior distillation or model-fusion methods mainly focus on model-level fusion. In contrast, FusionFactory jointly studies query-level, thought-level, and model-level fusion under the same benchmark and evaluation protocol.
> > >
> > > We have also revised **Section 3: FusionFactory** to connect this matrix to a stage-aware taxonomy, as illustrated in **Figure 4**. Query-level fusion is an early-stage intervention that selects a model before generation. Thought-level fusion is a mid-stage intervention that retrieves and reuses abstract reasoning templates during generation. Model-level fusion is a late-stage intervention that consolidates high-quality logged responses into a single model through supervised fine-tuning. This revised explanation makes clear that the matrix is not an arbitrary engineering checklist, but a structured organization of fusion methods according to where they intervene in the LLM pipeline.
> > >
> > > In addition, we have revised **Section 7** to better explain why studying the matrix leads to insights that cannot be obtained by studying only routing or only distillation. As shown in the cross-level comparison, different fusion stages exhibit different strengths and failure modes: thought-level fusion achieves the strongest overall performance, query-level fusion offers a favorable accuracy-cost trade-off, and model-level fusion may suffer from generalization limits. These findings demonstrate the value of comparing fusion levels under a shared benchmark, shared log source, and shared evaluation setup.
> > >
> > > Finally, we have added a discussion in **Limitations and Future Work** to further clarify the role of this matrix. The fusion-stage matrix provides a practical decision framework for users with different deployment constraints. Query-level fusion is suitable for lightweight and cost-sensitive serving, thought-level fusion is useful when users want to reuse reasoning patterns without fine-tuning, and model-level fusion is useful when users want to deploy a single fused model after offline training. Future work can further extend this matrix by studying hybrid or adaptive strategies that dynamically switch among fusion levels based on query difficulty, budget constraints, domain type, or uncertainty estimates.

---

> > > > ### Author Response · Authors · 2026-05-20
> > > > **Response (4)**
> > > >
> > > > **Q4. The limitation section is superficial. Privacy risks from log data, negative effects of low-quality logs, and failure cases in long texts / multi-turn dialogue, or other conditions are not discussed.**
> > > >
> > > > **Response:**  Thank you for the important comment. We agree that a more comprehensive discussion of limitations helps clarify the scope and responsible use of multi-LLM log data. We would like to clarify that the original paper already discusses several empirical limitations of FusionFactory, including the weaker performance of model-level fusion, the difficulty of fusing World Knowledge and Math domains, and the potential degradation caused by noisy or logically inconsistent fused information. However, we agree that the previous limitation discussion did not sufficiently cover broader risks and failure conditions associated with log-based fusion.
> > > >
> > > > To address this concern, we have expanded the **Limitations and Future Work** discussion to cover three additional aspects.
> > > >
> > > > First, we added a discussion of privacy risks from multi-LLM log data. Since log data may contain user queries, model responses, and potentially sensitive interaction traces, directly storing or reusing raw logs can raise privacy and data-governance concerns. We now clarify that our benchmark is constructed from public evaluation datasets and does not involve private user data. We also discuss that future real-world deployment of log-based fusion should incorporate privacy-preserving mechanisms, such as data anonymization, filtering of personally identifiable information, access control, retention policies, and possibly aggregation or synthetic-log generation before logs are used for training, retrieval, or distillation.
> > > >
> > > > Second, we added a discussion of the negative effects of low-quality logs. FusionFactory relies on logged responses and feedback signals, so low-quality, biased, hallucinated, or incorrectly scored logs may propagate errors into routing, thought-template retrieval, or model-level fine-tuning. We now explicitly discuss how such noise can affect different fusion levels: query-level fusion may learn suboptimal routing decisions, thought-level fusion may retrieve misleading reasoning templates, and model-level fusion may amplify noisy or biased responses through supervised fine-tuning. This also helps explain some empirical failure cases observed in our cross-level comparison, such as the weaker behavior of fusion on knowledge-intensive or strict-reasoning domains.
> > > >
> > > > Third, we added a discussion of failure cases under long-context and multi-turn settings. Our current benchmark mainly focuses on single-turn benchmark queries, while real-world logs often contain long documents, evolving user intents, tool-use traces, and multi-turn interaction histories. In these settings, simple query similarity may fail to retrieve the right historical examples, thought templates may omit important context, and model-level distillation may compress incomplete or stale interaction states. We now explicitly identify long-text reasoning, multi-turn dialogue, evolving user preferences, domain shift, and distribution shift as important conditions where the current version of FusionFactory may require additional mechanisms.
> > > >
> > > > Overall, these revisions make the limitation discussion more concrete and balanced. They clarify that multi-LLM log data is a promising supervision source, but its safe and effective use depends on privacy-aware data handling, log-quality control, and extensions to more complex interaction settings such as long-context and multi-turn dialogue.

---

> > > > > ### Author Response · Authors · 2026-05-20
> > > > > **Response (5)**
> > > > >
> > > > > **Q5. Some notations in the reward function are not clearly explained. Some figures and tables are overcrowded and hard to read.**
> > > > >
> > > > > **Response:**
> > > > > Thank you for pointing this out. We agree that clearer notation and improved visual readability can make the method easier to follow. We have revised the description of the reward function in **Section 4: Query-level Fusion** to make all variables and terms explicit.
> > > > >
> > > > > Specifically, in the original version, the reward function was written with abbreviated notation, which may make it unclear what each component refers to. In the revised version, we explicitly define the reward for a query-task pair and a selected LLM configuration as:
> > > > >
> > > > > `Reward(q, t, a) = alpha * P(q, t, a) - beta * C(q, t, a) + gamma * J(q, t, a)`
> > > > >
> > > > > where `q` denotes the query, `t` denotes the task or domain, and `a` denotes the router action, namely the selected LLM or LLM-response-mode configuration. `P(q, t, a)` denotes the task-specific performance score, `C(q, t, a)` denotes the inference cost estimated from token consumption and API pricing, and `J(q, t, a)` denotes the LLM-judge score. The coefficients `alpha`, `beta`, and `gamma` control the relative importance of task performance, cost, and judge-based response quality. This revision clarifies the meaning of the reward function and how it is connected to the structured log fields used in query-level fusion.
> > > > >
> > > > > For the issue of overcrowded figures and tables, we have also revised the presentation to improve readability. In particular, we adjusted the sizes and font scales of the relevant figures and tables, reduced visual clutter where possible, and improved spacing to make the key information easier to read. These changes mainly affect the benchmark construction figure, the method overview figure, and the main comparison tables. The revised figures and tables now better highlight the core message while keeping detailed information accessible in the text and appendix.
> > > > >
> > > > > Overall, these revisions improve the clarity of the reward-function notation and enhance the readability of the visual presentation without changing the underlying method or experimental conclusions.

---

### Author Response · Authors · 2026-05-20
**Global Response**

We sincerely apologize for the delayed submission of our rebuttal. Due to temporary constraints on laboratory servers and shared computational resources, completing the additional experiments and revisions required more time than expected. We appreciate the AE’s understanding and sincerely thank the AE and reviewers for their patience, careful reading, and constructive feedback.

In response to the reviewers’ comments, we revised the manuscript in four main aspects: clarifying the formulation and positioning of multi-LLM log data, strengthening the motivation and taxonomy of the three fusion levels, adding targeted experiments and comparisons, and improving notation, limitations, figures, tables, and overall presentation. All manuscript revisions are highlighted in blue.

---

## Summary of Main Concerns and Responses

| Dimension | Key Concerns | Our Main Actions | Status in Revision |
|---|---|---|---|
| **Contribution & Positioning** | What is the unique contribution beyond existing routing, distillation, and API-compatible methods? | Clarified that our main contribution is a unified benchmark and framework for studying how multi-LLM log data can support query-level, thought-level, and model-level fusion. | The contribution and positioning are now clearer relative to prior work. |
| **Log Data Definition** | The definition of multi-LLM log data was ambiguous. | Added a formal definition of each log entry as `(q, t, m, s, r, c, y, j)`, including query, task, model, response mode, response, cost, performance, and LLM-judge score. | The log-data setting is now clearly distinguished from ordinary response collections. |
| **Fusion Taxonomy** | The three fusion levels lacked a clear organizing principle. | Reframed query-, thought-, and model-level fusion as interventions before generation, during generation, and after response collection. | The three levels are now presented as a stage-aware taxonomy. |
| **Experiments & Comparisons** | Reviewers requested stronger evidence and clearer aggregate statistics. | Added/clarified comparisons with the best individual LLM, near-duplicate removal, Self-Consistency, LLM-Blender, and FuseLLM-adapted. | The empirical evidence is broader and easier to assess. |
| **Efficiency & Practicality** | Cross-level cost comparison was not clear enough. | Clarified the cost trade-offs among query-level routing, thought-level retrieval, and model-level fine-tuning/inference. | The efficiency discussion is now more explicit and quantified. |
| **Limitations & Risks** | The limitation section was superficial. | Expanded discussion of privacy risks, noisy logs, low-quality supervision, long-context inputs, multi-turn dialogue, and transparency risks. | The limitations section is now more comprehensive and balanced. |
| **Notation & Presentation** | Some notation, figures, and tables were unclear or overcrowded. | Revised reward-function notation and improved figure/table sizes, fonts, spacing, and captions. | The manuscript is clearer and easier to read. |

---

## Overall Changes in the Revised Manuscript

- Clarified the definition and scope of multi-LLM log data.
- Strengthened the positioning of FusionFactory against prior routing, distillation, and response-fusion methods.
- Added targeted experiments and comparisons requested by reviewers.
- Rewrote several result interpretations using clearer aggregate statistics and more cautious wording.
- Expanded the limitation discussion to cover privacy, noisy logs, and complex real-world settings.
- Improved reward-function notation, figure readability, table formatting, and captions.

---

## Appreciation

We sincerely thank the AE and all reviewers for their thoughtful and constructive feedback. Their comments helped us improve the clarity, rigor, and completeness of the paper.

---

### Author Response · Authors · 2026-06-21
**Follow up on the review progress**

Dear Action Editor and Reviewers,

We hope you are doing well. We would like to kindly follow up on our revised submission, which was uploaded on May 20 after incorporating the reviewers’ requested changes. The revised content has been marked in blue for ease of checking.

We sincerely appreciate the reviewers’ constructive feedback and the Action Editor’s time in handling our submission. We understand that the review process takes time, but we would be grateful for any update or further feedback when available.

Please let us know if there are any remaining concerns or additional revisions we should address.

Thank you again for your time and consideration.

Best regards,

The Authors

---

### Decision · Action_Editor_THVd · 2026-07-01

**Recommendation:** Accept as is

**Audience:**

Yes

**Audience Explanation:**

The provided benchmark is a useful asset to the general audience interested in utilizing multi-LLM.

One concern regarding this point is that the models being used are not very new. However, I do not think this is a very critical drawback from the readers' side, especially for a journal paper.

**Claims And Evidence:**

Yes

**Claims Explanation:**

Reviewer AD3M has pointed to several parts that needs better substantiations of the claim. This has been addressed quite comprehensively by moving the focus to the benchmark from the method itself, clarifying the positioning of the works, and added experiments.

The key concern that remains may be about the novelty of the algorithm---or alternatively, how well the authors revised the manuscript to prevent readers from believing that the algorithm itself has much novelty. After reading the revised manuscript, I did not get any impression that the authors are making any strong claim about the novelty of the method itself.

---

> ### Author Response · Authors · 2026-07-01
>
> Dear Action Editor,
>
> Thank you very much for your careful handling of our submission and for the constructive summary. We sincerely appreciate the time and effort from you and the reviewers throughout the revision process. We have now submitted the final camera-ready version of the manuscript. Thank you again for your guidance and consideration.
>
> Best regards,
>
> Authors